# Multivalent interactions facilitate motor-dependent protein accumulation at growing microtubule plus-ends

Renu Maan[1,5], Louis Reese[1,2,5], Vladimir A. Volkov[1,2,3,5], Matthew R. King[2,4], Eli O. van der Sluis[1], Nemo Andrea[1], Wiel H. Evers[1], Arjen J. Jakobi[1] & Marileen Dogterom ● [1,2] ✉

Growing microtubule ends organize end-tracking proteins into comets of mixed composition. Here using a reconstituted fission yeast system consisting of end-binding protein Mal3, kinesin Tea2 and cargo Tip1, we found that these proteins can be driven into liquid-phase droplets both in solution and at microtubule ends under crowding conditions. In the absence of crowding agents, cryo-electron tomography revealed that motor-dependent comets consist of disordered networks where multivalent interactions may facilitate non-stoichiometric accumulation of cargo Tip1. We found that two disordered protein regions in Mal3 are required for the formation of droplets and motor-dependent accumulation of Tip1, while autonomous Mal3 comet formation requires only one of them. Using theoretical modelling, we explore possible mechanisms by which motor activity and multivalent interactions may lead to the observed enrichment of Tip1 at microtubule ends. We conclude that microtubule ends may act as platforms where multivalent interactions condense microtubule-associated proteins into large multi-protein complexes.

Growing microtubule plus-ends recruit an evolutionary conserved network of proteins interacting with end-binding (EB) proteins[1]. This network exists as a multivalent protein assembly that recognizes features of growing microtubule ends, such as GTP hydrolysis intermediates[2], bent tubulin protofilaments[3] and tubulin interfaces that are unavailable on closed microtubules[4]. In fungi, the microtubule plus-end tracking (MPET) system is crucial to establish cell polarity by asymmetrically transporting polarity markers to the cellular cortex[5,6]. Once associated with the cellular cortex, many of these markers behave like clusters[7], which raises the question whether clusters may already be formed at growing microtubule ends before being deposited at the cortex[8].

A minimal protein network for MPET was first reconstituted in vitro using purified proteins from *Schizosaccharomyces pombe*[9].

The three proteins that are necessary and sufficient for successful in vitro plus-end tracking are Mal3 (EB homologue), Tea2 (kinesin-7 homologue) and Tip1 (CLIP-170 homologue). Accumulation of Tip1 and Tea2 at the microtubule end is Mal3-dependent both in vitro and in vivo[5,10]. Mal3 is needed for ATPase activity and processive transport of Tea2 (ref. [11]). However, affinity of Mal3 for microtubules is independent of Tea2 and Tip1. Tip1 has been shown to interact with the EB homology domain of Mal3 through its CAP-Gly domain[10], as also shown for Tip1 homologue CLIP-170 and other plus-end tracking proteins (+TIPs) interacting with EB proteins[1,12–14]. Tea2 interacts with Mal3 through its N-terminal extension and with Tip1 through its coiled-coil region[6,11,15]. As many of these interactions happen through disordered protein regions (Extended Data Fig. 1a), we hypothesize that the Mal3/Tip1/

[1]Department of Bionanoscience, Kavli Institute of Nanoscience, Delft University of Technology, Delft, the Netherlands. [2]Physiology Course 2017, Marine Biological Laboratory, Woods Hole, MA, USA. [3]Present address: School of Biological and Behavioural Sciences, Queen Mary University of London, London, UK. [4]Present address: Department of Biomedical Engineering, Washington University in St. Louis, St Louis, MO, USA. [5]These authors contributed equally: Renu Maan, Louis Reese, and Vladimir A. Volkov. ✉e-mail: m.dogterom@tudelft.nl

Tea2 network may be formed by multivalent low-affinity interactions that are a hallmark of liquid–liquid phase separation (LLPS)[16,17].

LLPS is the phenomenon of reversible de-mixing of miscible components from their homogeneous mixture driven by microscopic interactions between the molecules[18]. Eukaryotic cells contain many membrane-bound and membrane-less organelles that form through similar phase separation processes. Examples include Cajal bodies, nuclear speckles, nucleoli, stress granules and P-bodies[19–21]. Recently, a number of microtubule-associated proteins have been reported to undergo similar de-mixing in vitro with proposed relevance for microtubule dynamics, nucleation, branching and so on[22–26]. Note, however, that the importance of these liquid- and gel-like assemblies for cellular function is still controversial[16,27]. Also, while it is widely accepted that disordered protein regions often drive interactions leading to phase separation[17], it should be noted that some of the phenomena explained through phase separation of disordered regions could be interpreted as being produced by site-specific interactions as well[28].

In this Article, we investigate the role of multivalent interactions in the formation of comets of fission yeast MPET proteins at growing microtubule ends in an in vitro reconstitution experiment. Using a combination of fluorescence microscopy, electron cryo-tomography (cryo-ET) and protein truncation, we study the formation of both phase-separated droplets and comets under crowding and non-crowding conditions, focusing on the contribution of intrinsically disordered regions (IDRs) in the Mal3 protein. We conclude that multivalent interactions contribute to a network-like architecture of plus-end comets, forming disordered structures that are easily driven into phase-separated dense droplets under crowding conditions. We propose that these non-stoichiometric structures allow for the efficient motor-driven accumulation of Tip1 at microtubule ends. We finally use stochastic modelling of motor-driven cargo transport to explore how multivalent interactions may enhance this accumulation at microtubule ends.

## Results

### MPET proteins form a complex on microtubule lattice and ends

We reconstituted the fission yeast MPET network in vitro using bacterially expressed proteins Mal3, Tea2 and Tip1, as reported previously[9] (Fig. 1a). Using total internal reflection fluorescence (TIRF) microscopy and double labelling of either Mal3-Alexa647 and Tip1:GFP (Fig. 1b) or Mal3-Alexa488 and Tea2-Alexa647 (Fig. 1c), we observed that all three proteins were transported on the microtubule lattice in the direction of the microtubule plus-end and were all present in an end-tracking comet, confirming that they form a complex. As all three proteins contain disordered, low-complexity regions (Extended Data Fig. 1a), we hypothesized that efficient plus-end accumulation of the Mal3/Tea2/Tip1 protein network is facilitated through multivalent or non-stoichiometric protein interactions. To test this hypothesis, we investigated the behaviour of the protein network under crowding conditions, first without and then with microtubules.

### Mal3, Tea2 and Tip1 co-condense under crowding conditions

As Mal3 is an autonomous end-tracker and also plays a key role in motor activation needed for plus-end tracking of the MPET network[9], we first focused on the ability of Mal3 to form condensates. At high concentrations, Mal3 readily formed condensates in the presence of polyethylene glycol (PEG) 6k (Fig. 1d) that fused together like fluid droplets (Supplementary Video 1). To probe the robustness of droplet formation, we systematically explored the effects of Mal3 and PEG concentration as well as PEG chain length. At 200 nM Mal3:mCherry, a typical concentration in microtubule end-tracking assays, and 5% (w/v) of PEG-35k, Mal3 produced robust protein droplets (Extended Data Fig. 1b). In the additional presence of 20 nM Tea2-Alexa647 and 150 nM Tip1:GFP, typical concentrations for microtubule end-tracking

reconstitutions, we observed co-localization of Tea2 and Tip1 with Mal3 condensates (Fig. 1e). Also, Tea2 and Tip1 formed condensates under similar crowding conditions and concentrations on their own (Extended Data Fig. 5).

### Mal3, Tea2 and Tip1 co-condense on microtubules

In the presence of dynamic microtubules growing from coverslip-anchored seeds and 5% PEG-35k, Mal3:GFP coated the entire microtubule lattice (Fig. 1f). When Tea2 and Tip1:GFP were added to the PEG-containing assay, we observed both motor traces at the lattice and bright comets at microtubule plus-ends (Fig. 1g). These plus-end-bound comets could transfer from the end of one microtubule to the lattice of another, spread out, be transported again towards the new plus-end and then merge with the comet of the second microtubule (Fig. 1h and Supplementary Video 2).

When only immobilized seeds but no soluble tubulin were present, we observed Mal3 binding to the GMPCPP seeds, contrary to non-crowding conditions where Mal3 did not interact with the seeds (Extended Data Fig. 1c). In the presence of all MPET proteins, we observed Tip1:GFP transport towards the plus-end on seeds. Presumably, PEG-assisted Mal3 binding to the seeds was sufficient to induce Tea2 activity and hence Tip1:GFP transport towards the plus-end. Droplets were observed to form at the plus-ends of the seeds that grew over time owing to continuous Tea2-driven transport along the seeds (Fig. 1i (top) and Supplementary Video 3). Finally, when seeds were not attached to the coverslips, and motors were non-specifically binding to the glass surface, seeds started gliding and depositing trails of droplets behind their plus-ends (Fig. 1i (bottom) and Supplementary Video 4), similar to the Plateau–Rayleigh instability[29,30].

Together, these observations provide evidence that, in the presence of crowding agent, Mal3, Tip1 and Tea2 together form condensates both in the absence and in the presence of microtubules. The observed condensates are liquid-like in nature, can coat the microtubule lattice and can be transported by Tea2 motors towards the plus-ends of microtubules.

### Cryo-ET of MPET protein droplets and comets

We next asked whether droplet-like comets also form in the absence of crowding agents. Given the small size of normal comets, we turned to cryo-ET for higher spatial resolution. We first added pre-formed droplets made by incubating the Mal3/Tip1/Tea2 mixture with 10% PEG-6k to holey carbon grids and vitrified them (Fig. 2a). In these conditions, we observed spherical droplets with fine internal grain (Fig. 2b,c). To prevent non-specific adsorption in experiments with tubulin, we adapted passivation methods previously established for treatment of glass coverslips[31]. We silanized a SiO film on the grids, adsorbed anti-DIG IgG to the silanized surface and then made the film hydrophilic by incubation with Pluronic F-127 (Fig. 2d). This treatment allowed us to firmly attach DIG-labelled GMPCPP seeds, while rejecting the binding of other proteins from solution. We then added tubulin in the presence or absence of Mal3 or the complete MPET network and plunge-froze the grids after 5–7 min of microtubule growth.

To facilitate analysis of microtubule end structures, we used cryoCARE, a neural network-driven denoising algorithm designed to increase the signal to noise ratio in individual tomograms[32] (for details, see Methods; Extended Data Fig. 2). Microtubule polarity was determined from moiré patterns or protofilament shapes in microtubule cross-sections[33,34] (Extended Data Fig. 3a and Supplementary Table 1). In the absence of end-tracking proteins, we observed microtubules growing with flared protofilaments at their ends, as described previously[35], and no lattice or end decoration (Fig. 2e). Adding Mal3 alone did not produce clearly visible densities at microtubule ends (Fig. 2e), but we observed a clear diffuse coating at the ends of growing microtubules when all MPET components were present (Fig. 2e and Extended Data Fig. 3).

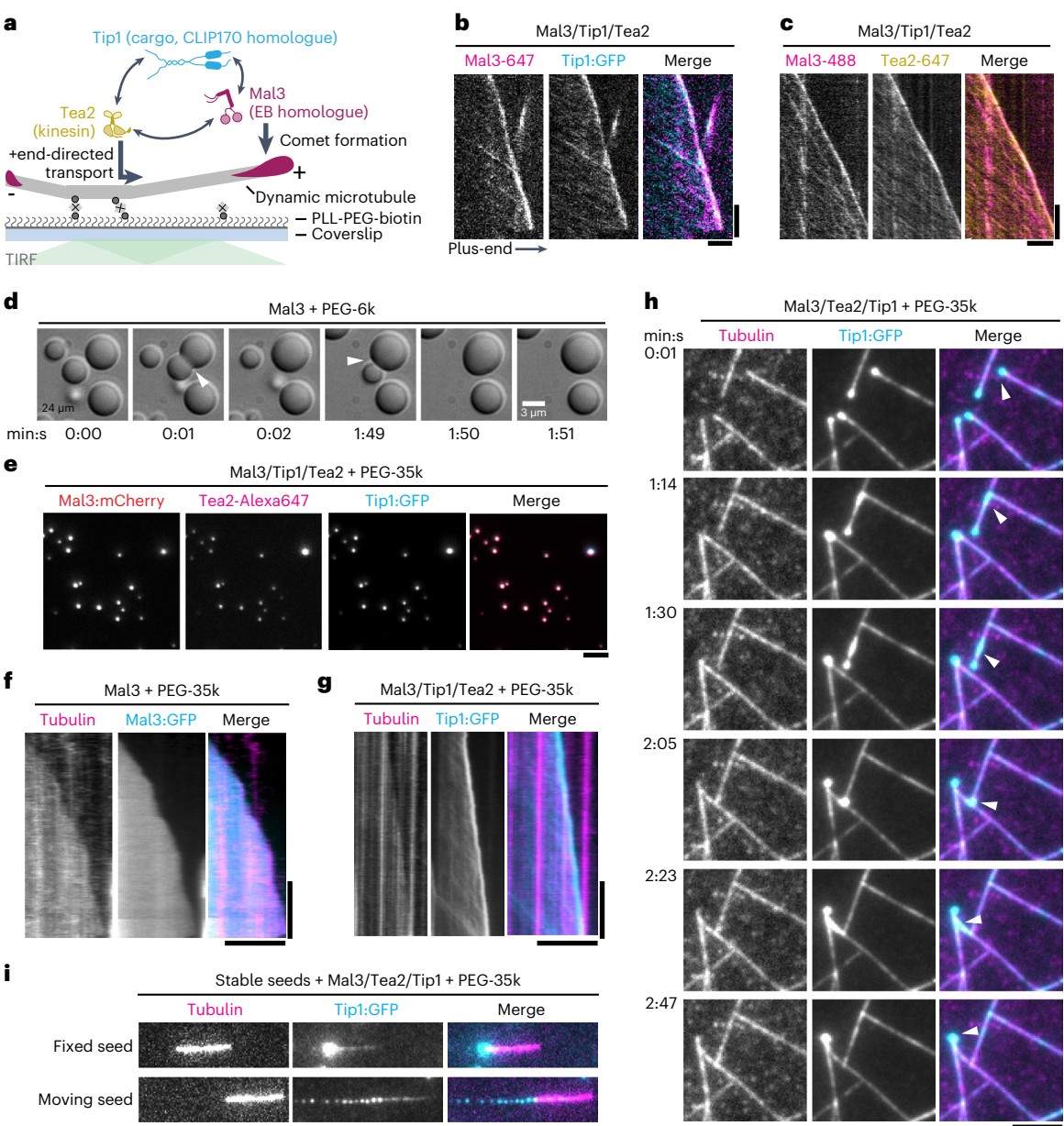

**Fig. 1 | Fission yeast microtubule end-tracking system undergoes phase separation under crowding conditions in vitro. a**, Cartoon showing the interactions among the three plus-end tracking proteins, Mal3, Tea2 and Tip1. **b**,**c**, Kymographs of MPET reconstitutions showing co-localization of Mal3-Alexa647 and Tip:GFP (**b**), and Mal3-Alexa488 and Tea2-Alexa647 (**c**) on both the microtubule lattice and the plus-end. Scale bars, 5 μm and 60 s. **d**, Large Mal3 condensates form in the presence of 10% (w/v) PEG-6k at 24 μM concentration. Two condensates can be seen fusing over time (arrowheads). **e**, Co-condensation of Mal3:mCherry (200 nM), Tea2-Alexa647 (20 nM) and Tip1:GFP (150 nM) in the presence of 5% PEG-35k at the protein concentrations used in the MPET reconstitution assays. **f**, Mal3:GFP (200 nM) coats the entire microtubule lattice in the presence of 5% PEG-35k while there is no distinct accumulation at

the microtubule end. **g**, Combination of Mal3 (200 nM) with Tea2 (20 nM) and Tip1:GTP (150 nM) in the presence of 5% PEG-35k leads to Tip1:GFP accumulation at the plus-end with motor traces visible on the lattice. **h**, A droplet of Mal3/Tea2/Tip1 formed at the plus-end of one microtubule gets transferred to the lattice of another (arrowhead). The transferred droplet spreads on the microtubule lattice and moves towards the plus-end where it fuses with the already existing Mal3/Tea2/Tip1 droplet. See also Supplementary Video 2. Scale bars, 5 μm. **i**, Top: MPET reconstitution on GMPCPP-stabilized seeds in the presence of 5% PEG-35k. Bottom: deposition of Mal3/Tea2/Tip1 droplets by the moving seed on the glass surface in the presence of 5% PEG-35k. Seed movement occurs through non-specific binding of Tea2 to the surface.

---

To assist the interpretation of the reconstructed tomograms, we used volume segmentation to highlight tubulin and microtubules (cyan) and non-tubulin densities (yellow) (Fig. 2f). Together with polarity assignment, this allowed us to visualize massive microtubule end-bound structures at the plus-ends in the presence of Mal3/Tea2/Tip1 (Fig. 2f, Extended Data Fig. 3b and Supplementary Video 5). Structures binding to minus-ends in the presence of Mal3/Tea2/Tip1

(Supplementary Video 6) to plus-ends in the presence of Mal3 alone (Extended Data Fig. 3c and Supplementary Video 7) or to plus-ends in the absence of additional proteins (Extended Data Fig. 3d and Supplementary Video 8) appeared much smaller. Interestingly, when we added Mal3/Tea2/Tip1 in the presence of PEG to microtubules pre-polymerized in the presence of Mal3/Tea2/Tip1 without PEG, we observed a subset of comets that looked similar to the ones we

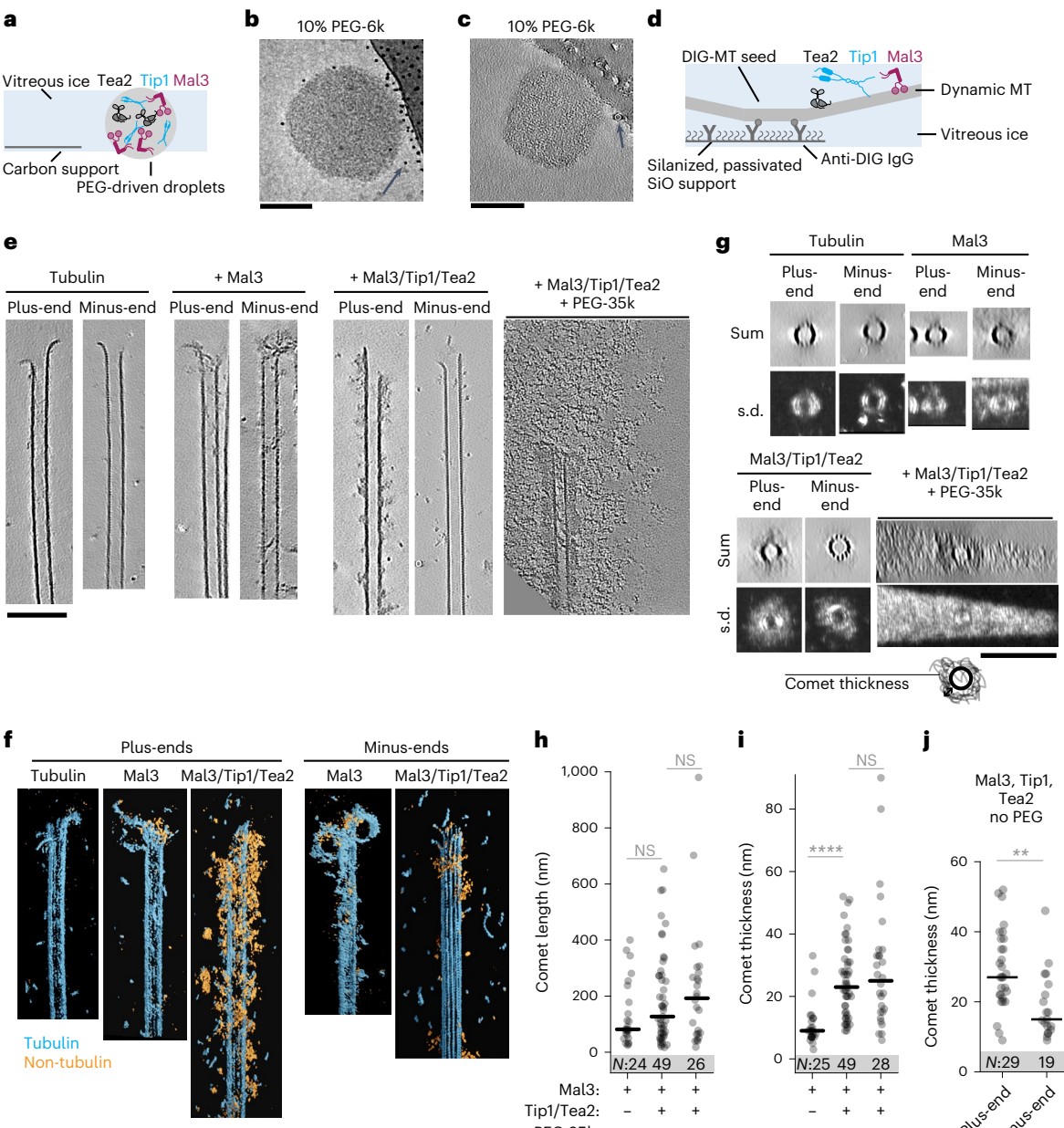

**Fig. 2 | Cryo-ET analysis of Mal3/Tip1/Tea2 assemblies. a**, Schematics of sample preparation for analysis of PEG-driven droplets. **b**, A single 10 s exposure of a droplet in a hole attached sideways to a carbon support. **c**, A 0.7-nm-thick slice through a 3D tomographic volume. Arrows in **b** and **c** show 5 nm gold beads added as fiducials for 3D reconstruction (note that the gold density is erased in the 3D volume, but not in the 2D image). **d**, Schematics of sample preparation to study microtubule-bound assemblies of Mal3/Tip1/Tea2. **e**, Slices (0.7 nm thick) through 3D tomographic volumes recorded in the conditions indicated (for more examples, see Extended Data Figs. 2 and 3). **f**, 3D renders of tomograms containing microtubule ends with bound material, segmented using *tomoseg* module of EMAN2.2 (for details, see Methods). Cyan shows tubulin and microtubules; orange indicates all non-tubulin densities. **g**, Projections along the microtubule length of volumes presented as sum of slices (top) and s.d.

(bottom). **h,i**, Comet length (**h**) and thickness (**i**) in the presence of Mal3 alone or Mal3/Tip1/Tea2 (with and without PEG). *P* values: Mal3 with or without Tip1/Tea2, 0.2226; Mal3/Tip1/Tea2 with or without PEG, 0.5973; Mal3 with or without Tip1/Tea2, 1·10-8 (****P = < 0.0001); Mal3/Tip1/Tea2 with or without PEG, 0.6948. NS, not significant. **j**, Thickness of comets formed by Mal3, Tip1 and Tea2 in the absence of PEG on plus- or minus-ends of dynamic microtubules. *P* value: 0.012 (**P = < 0.01). Each datapoint represents a single microtubule end. Lines show median, and numbers in the shaded area (*N*) show number of analysed microtubule ends (pooled across two independent experiments per condition). *P* values are reported according to the two-tailed Mann–Whitney test. Grid preparations were repeated three times; images and analysis is shown from two independently prepared grids for each condition. Scale bars, 100 nm. Numerical data are available in source data.

observed in the absence of PEG, and a subset that were surrounded by diffuse material extending over hundreds of nanometres from the microtubule walls (Fig. 2e and Extended Data Fig. 3e).

We further analysed microtubule cross-sections to obtain quantitative information on the microtubule end-bound structures (Fig. 2g,h). The average thickness of comets extending outwards from the microtubule surface in the presence of Mal3 alone was 11 ± 7 nm (here and onwards mean ± standard deviation (s.d.)), considerably thinner than 25 ± 12 nm in the presence of Mal3, Tip1 and Tea2 (Fig. 2i). The presence of PEG did not result in a statistically significant difference in comet length or thickness (Fig. 2h,i) (not taking into account the diffuse material surrounding the comets). The differences in comet

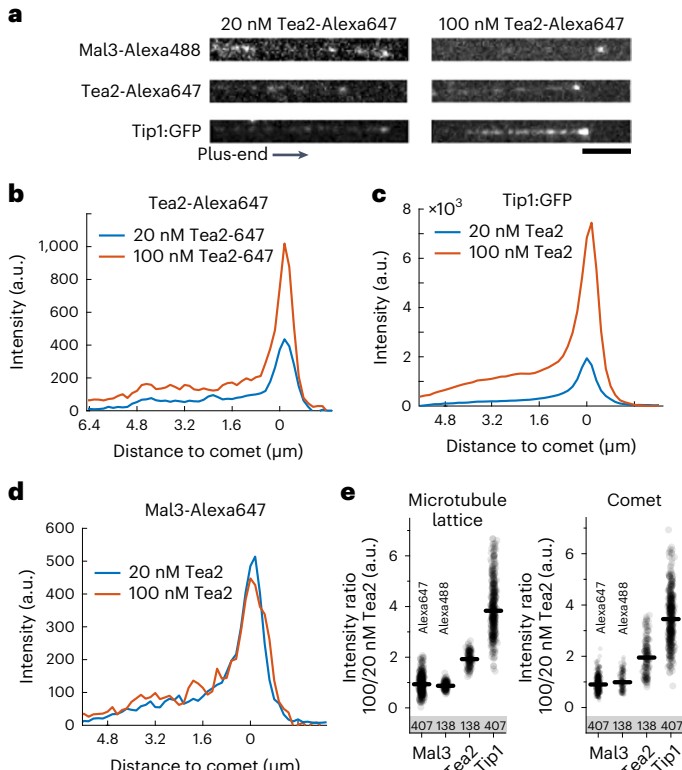

**Fig. 3 | Non-stoichiometric accumulation of Tip1 at microtubule plus-ends.**
**a**, Examples of individual Mal3, Tea2 and Tip1 intensity profiles at two different
concentrations of Tea2 (20 nM and 100 nM). **b**, Averaged Tea2-Alexa647 intensity
profiles. **c**, Averaged Tip1:GFP intensity profiles. **d**, Averaged Mal3-Alexa647
intensity profiles. Data in **c** and **d** were extracted from the same experiment for
each Tea2 concentration, whereas Tea2-Alexa647 data shown in **b** were recorded
in two separate experiments that contained dark Tip1 and Mal3-Alexa488.
Profiles were averaged for microtubules of similar length. See also Extended
Data Fig. 4. **e**, Ratio of lattice (left) and comet (right) intensities between 100 nM
and 20 nM Tea2 for microtubules of all lengths. The total number of observed
intensity profiles at 100 nM is indicated for each condition. Numerical data are
available in source data.

length were not significant (Fig. 2h). Plus-ends carried thicker comets
of Mal3/Tip1/Tea2 ($29 \pm 11$ nm) compared with minus-ends in the same
sample ($19 \pm 9$ nm) (Fig. 2j, right). The polarity-dependent thickness
of comets is consistent with the plus-end-directed motility of Tea2
bringing its cargo, Tip1, to the plus-ends of microtubules.

There is clearly a difference between the shape and the internal
organization of the spherical PEG-driven droplets shown in Fig. 2b,c and
the microtubule-bound comets that appear as more loosely structured.
Interestingly, when PEG together with Mal3/Tip1/Tea2 was added to
comets pre-formed in the absence of PEG, a separate comet structure
remained visible even when surrounded by diffuse material, hinting
that these structures were formed in different ways. Yet, it is possible
that multivalent interactions responsible for the formation of droplets
under crowding conditions are also facilitating the formation of the
network-like architecture of motor-driven plus-end comets observed
in cryo-ET.

#### Non-stoichiometric Tip1 accumulation at microtubule ends
The network-like architecture described above may facilitate the
non-stoichiometric accumulation of Tip1 cargo at microtubule ends.
To address this hypothesis, we measured fluorescence intensities of
MPET proteins along the microtubules at two Tea2 concentrations
(20 nM and 100 nM; Fig. 3a). The average intensity profiles of Tea2[Alexa647],

Tip1:GFP and Mal3-Alexa647 demonstrated a similar, specific shape:
a shallow intensity increase starting at the microtubule seed, a con-
stant average intensity along the microtubule lattice and a peak at
the microtubule plus-end (Fig. 3b–d). Interestingly, at higher motor
concentration, Tip1 intensity increased more than Tea2 intensity itself
both on the microtubule lattice and at microtubule ends (Fig. 3b,c).
In contrast, the intensity of Mal3-Alexa647 did not change with Tea2
concentration (Fig. 3d).

We summarized the effect of Tea2 concentration on the end
accumulation of Mal3 and Tip1 by calculating the ratios of intensities
between the two Tea2 concentrations for Mal3 and Tip1 on the micro-
tubule lattice and in the comet (Fig. 3e). An increase in Tea2 concen-
tration had no influence on the amount of labelled Mal3 protein that
localized at the microtubule plus-end. On the other hand, Tip1:GFP
localization to the plus-end was disproportionately affected by Tea2
concentration. An increase from 20 nM to 100 nM Tea2 led to a roughly
fourfold increase of Tip1:GFP intensity at the plus-end, whereas the
Tea2 intensity itself was increased by only a factor of 2. Apparently, the
amount of Tip1 that is present on the microtubule does not follow the
density of motor proteins on the microtubule in a stoichiometric way.
In fact, we show that the presence of Tip1 responds in a non-linear way
to the concentration of Tea2 over a range of concentrations (Extended
Data Fig. 4), in agreement with previous observations[8]. Note also that
there is large variability in the Tip1 intensity between individual micro-
tubules (Extended Data Fig. 4a), which we interpret as another sign
that the accumulation of the cargo Tip1 is not limited by one-on-one
interactions with motor proteins.

#### Distinct domains of Mal3 drive formation of comets and LLPS
Having established that our three-component network is capable of
both droplet formation and non-stoichiometric protein accumula-
tion at microtubule ends, we set out to elucidate the contributions
of disordered protein regions to both comet formation and LLPS. As
Mal3 is central to comet formation of all three proteins, we studied
different truncations of Mal3. Full-length Mal3 contains two folded
domains: a calponin-homology (CH) domain and an EB-homology
domain (EB HD), and two IDRs: IDR1, which connects the CH domain
to EB HD, and the C-terminal IDR2 (Fig. 4a). Note that the C-terminal
IDR2 domain is not present in Mal3's homologue EB1, which contains
a much shorter negatively charged C-terminal tail[36]. We first focused
on dissecting the contributions of these domains to formation of Mal3
comets on microtubule ends without Tea2 or Tip1, and in the absence
of crowding conditions (Fig. 4b,c).

In the absence of Tea2 and Tip1, 200 nM of full length Mal3:GFP
coated the entire microtubule lattice without a clear saturation at the
plus-end, in contrast to the full MPET network (compare Figs. 4c (top
left) and 1b, respectively). Mal3-ΔIDR2 showed a lower binding affinity
to the microtubule lattice than full-length Mal3, and formed slightly
brighter comets at the microtubule ends (Fig. 4c,d). We did not observe
any comet formation or lattice binding with Mal3 mutants containing
only the CH domain (Fig. 4b,c). Other Mal3 mutants were binding
mostly to the growing end, rather than the microtubule lattice, and
formed comets with very low intensity (Fig. 4d). We conclude that, in
addition to previously described microtubule binding through the CH
domain and the role of dimerization through the EB HD[37,38], IDR1 also
contributes to efficient comet formation by Mal3. In contrast, IDR2
appears not to contribute to Mal3's affinity to the microtubule end but
only to its affinity to the lattice (potentially via Mal3 self-interactions;
see below). Note that in previous work on EB1 truncations, it was
observed that removal of the C-terminal tail (where IDR2 is located in
Mal3) led to stronger instead of weaker lattice binding[36]. This effect
was attributed to the removal of a short negatively charged section of
the protein, which is expected to destabilize electrostatic interactions
with the positively charged microtubule lattice. While it is difficult to
disentangle the effect of charge from the contribution of multivalent

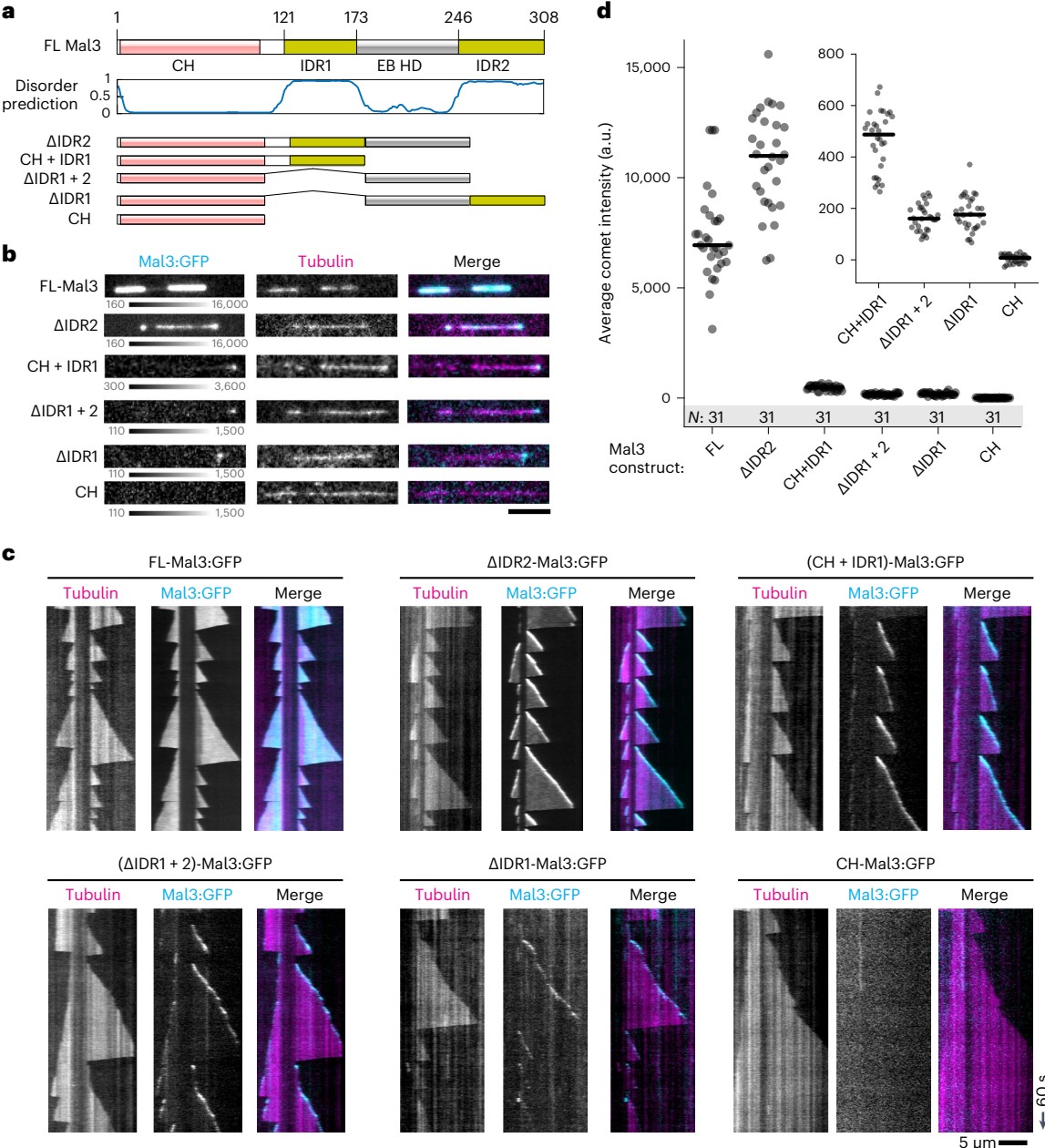

**Fig. 4 | Mal3 IDR1 and EB HD are required for comet formation. a**, Pictorial representation of the full length Mal3 together with disorder prediction[59] and the truncations used in this study. **b**, FL-Mal3 and Mal3 truncation mutants (200 nM) decorating the growing microtubule ends. **c**, Kymographs for the end-tracking experiments presented in **b**. **d**, Average comet intensity for full-length Mal3 and Mal3 truncations. Lines show median, and numbers in the shaded area (*N*) show number of microtubule ends analysed pooled from two independent experiments for each Mal3 truncation. Inset shows scaled-up graph for four truncations with poorest binding. Scale bars, 5 μm and 60 s. Numerical data are available in source data.

interactions, it should be noted that, unlike Mal3, EB1 does not have a sizeable, disordered region at its C-terminal end.

We next wondered which domains of Mal3 were important for the protein's self-interactions under crowding conditions. When Mal3 mutants were incubated at a concentration 1 μM with 5% PEG-35k, we observed that domain deletions preventing comet formation on microtubules also prevented droplet formation (Fig. 5a). In addition, Mal3-ΔIDR2, which reduced microtubule lattice- but not microtubule end-binding, also formed smaller condensates than the full-length protein in the presence of a crowding agent. We conclude that IDR1 and EB HD are necessary both for Mal3 self-interactions and for Mal3 interaction with the microtubule end, while IDR2 is necessary for Mal3

self-interactions and interaction with the microtubule lattice, but not the microtubule end (Figs. 4b,c and 5a).

### Roles of Mal3 domains in droplet and comet formation

To pin-point the interactions between Mal3, Tea2 and Tip1 under crowding conditions, we next designed a scaffold-client assay (Fig. 5b,c and Extended Data Fig. 5). Scaffold condensates were formed by either Mal3, Tea2-Alexa647 or Tip1 by incubation with 10% PEG-6k, and 2 nM Mal3:GFP was added as a client. When non-fluorescent Mal3 and Tip1 were used, we additionally added 2 nM full-length Mal3:mCherry as a tag to visualize the scaffold condensates independent of Mal3:GFP construct localization. Figure 5b shows the outcome of a typical

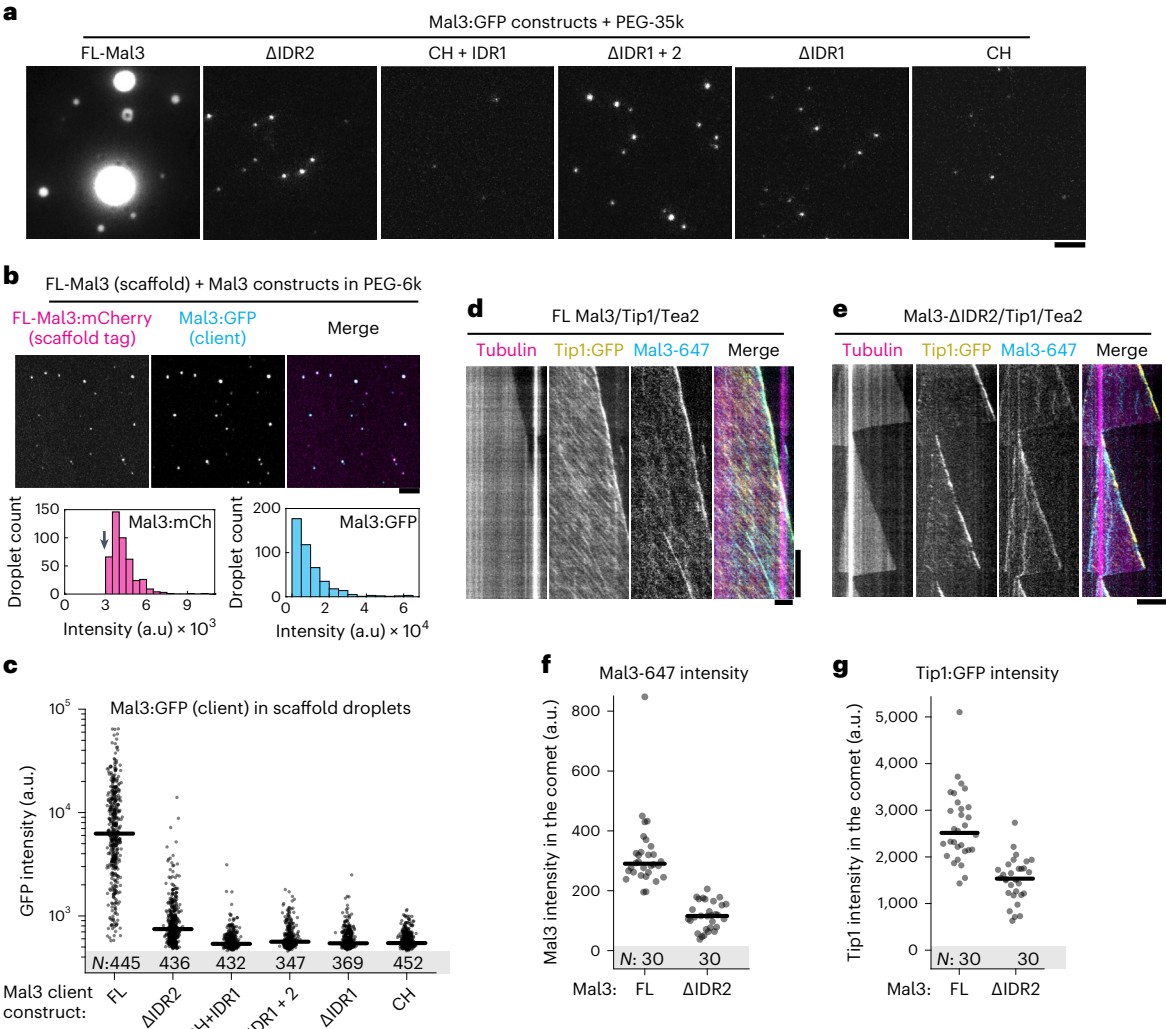

**Fig. 5 | Distinct domains of Mal3 mediate comet formation and LLPS.**
**a**, Condensates formed by full-length Mal3 and Mal3 truncations (1 µM) in the presence of 5% PEG-35k. **b**, Unlabelled FLMal3 (200 nM, scaffold) tagged with FL-Mal3:mCherry (2 nM, tag) was allowed to recruit Mal3:GFP (2 nM, client) in the presence of PEG-6k. Graphs show distributions of tag and scaffold Mal3 intensities (arrow denotes the threshold applied for speckle detection in images). **c**, Intensity of Mal3:GFP (full-length or truncated constructs) recruited to FL-Mal3 scaffold droplets. Lines show median, and numbers in the shaded area (*N*) show number of droplets analysed (one experiment per construct). **d,e**, Kymographs showing end-tracking by Mal3/Tip1/Tea2 in the presence of full-length Mal3 (**d**) or Mal3-ΔIDR2 (**e**) and in the absence of PEG. **f,g**, Intensities of Mal3-Alexa647 (**f**) and Tip1:GFP (**g**) in the comets forming in the presence of FL-Mal3 or Mal3-ΔIDR2. Lines show median, and numbers in the shaded area (*N*) show number of comets analysed (pooled across two independent experiments per condition). Numerical data are available in source data.

experiment, with non-fluorescent Mal3 as the scaffold (tagged with Mal3:mCherry), and full-length Mal3:GFP as a client. Deletion of any disordered region from Mal3:GFP prevented its recruitment to the Mal3 scaffold (Fig. 5c), reinforcing our conclusion that both IDR1 and IDR2 are important for Mal3–Mal3 interactions in crowding conditions.

We observed a direct interaction between full-length Tea2 and Mal3 in crowding conditions in the absence of Tip1 (Extended Data Fig. 5a). However, Mal3 constructs lacking IDR1 or IDR2 were recruited poorly to Tea2-Alexa647 scaffold (Extended Data Fig. 5a). Deletion of EB HD further disrupted recruitment of Mal3 to the Tea2 scaffold. These data indicate that crowding conditions strengthen Tea2–Mal3 interactions and that these interactions rely on the disordered regions in Mal3 as well as the EB HD. Finally, we used unlabelled Tip1 as the scaffold (Extended Data Fig. 5b) and Mal3:GFP truncations as the client. We again observed that Tip1 condensates predominantly recruited full-length Mal3:GFP, and to a much lesser extent Mal3-ΔIDR2, but failed to recruit the Mal3 constructs lacking the EB homology domain or IDR1.

We finally set out to correlate the recruitment behaviour observed in the scaffold-client assays with the capacity of truncated Mal3 constructs to couple Tip1/Tea2 transport to plus-end tracking on dynamic microtubules. Using Tip1:GFP fluorescence as a readout, we observed that Mal3 constructs lacking either IDR1, EB homology domain or both failed to recruit Tip1:GFP to microtubules altogether (Extended Data Fig. 6). Mal3-ΔIDR2 was still able to support Tip1 localization at microtubule ends, but unlike full-length Mal3, it did not co-localize with Tea2/Tip1 transported along the microtubule lattice (Fig. 5d,e). Furthermore, the intensity of both Mal3-ΔIDR2 and Tip1 in the comets was reduced compared with full-length Mal3 (Fig. 5f,g). Together, the analysis of Mal3 truncations leads us to conclude that robust three-component comets are formed by a combination of different molecular mechanisms. Mal3 interaction with itself, Mal3 interaction with the microtubule lattice, as well as Mal3 co-localization with motor tracks requires each of Mal3's IDRs. The formation of Mal3/Tip1 comets requires only Mal3 IDR1 and EB HD, but the additional presence of IDR2 enhances

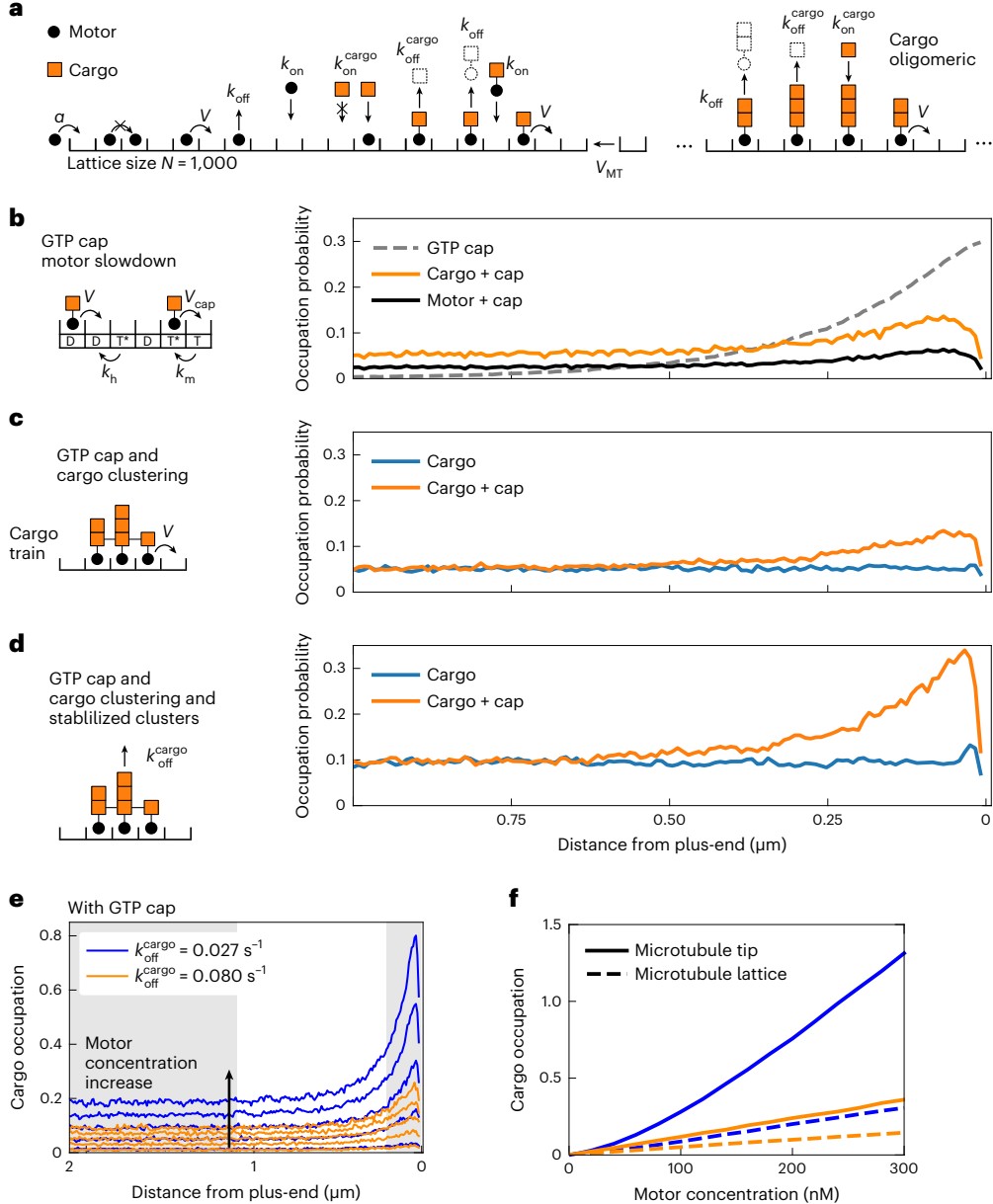

**Fig. 6 | Theoretical models and stochastic simulations. a**, Cartoon of a lattice model for motor transport with cargo oligomerization. Reactions include motor movement, attachment, detachment, a growing lattice, the binding and unbinding of cargo to bound motors, and a cargo multi-layer absorption/desorption process. Model parameters are provided in Supplementary Table 2. **b**, Nucleotide-dependent motor slowdown was implemented by including a GTP cap and two-step hydrolysis. The GTP profile decays exponentially (dashed line, normalized to 0.3) and the amounts of motors and cargos at the lattice end increase. **c**, Interactions between cargo particles was implemented such that neighbouring cargo particles form a cargo train that induces coherent movement of cargo-motor clusters. Cargo clustering does not lead to increasing amounts of cargo at the lattice end unless a GTP cap is implemented as in **b**. **d**, The stability of cargo clusters was increased by dynamically enhancing the motor dwell time in cargo clusters. This mechanism alone did not result in substantial accumulation of cargo, similar to **c**. However, cargo accumulation increased sharply in combination with a GTP cap. Motor concentration in **b**–**d** corresponds to ~100 nM Tea2 in experiments. **e**, Density profiles of stabilized cargo clusters (blue lines) and independent cargo clusters (orange lines) for a range of motor concentrations between 20 nM and 180 nM. **f**, The average cargo occupation is shown depending on motor concentration for the microtubule lattice (shaded area in **e** on the left; dashed lines) and the microtubule tip (shaded area in **e** on the right corresponding to ~200 nm; solid lines). Numerical data are available in source data.

the motor-dependent accumulation of Tip1 at growing microtubule ends. It thus appears that Mal3 self-interactions are needed to promote non-stoichiometric Tea2/Tip1 transport on the microtubule lattice.

**Theoretical models for motor-driven end accumulation**

To help understand the possible contribution of protein self-interactions to efficient motor-driven end accumulation, we turned to stochastic simulations, complementing a series of previously published models of single-component traffic jams[39]. It should be stressed that these simulations were not designed to exactly reproduce our experimental situation, which is highly complex: plus-end accumulation of the three MPET network components (Mal3, Tea2 and Tip1) is a result of both motor-driven transport towards the plus-end and autonomous inter-action of Mal3 with the comet region near the growing MT ends. Varying

the concentration of each of the components is likely to change the balance of complex formation both in solution and on the microtubules, complicating straightforward predictions about the resulting effects on both lattice coverage and end accumulation. The phenomenology of motor transport in the absence of cargo is well known[40,41], and because the binding/unbinding of cargo is an equilibrium process, it is not expected that simple 1:1 cargo binding changes any characteristic of these models. We therefore specifically focused on the possible effects of cargo clustering due to protein self-interactions.

A microtubule was represented as a growing one-dimensional lattice[42,43], and the motors as particles binding to and unbinding from the lattice and hopping towards the plus-end[39,44,45], where each lattice site can be occupied by only one motor (Fig. 6a). The cargo was represented by a second set of particles that bind to and unbind from the motor particles. Mal3 was not simulated explicitly, because in our experiments Mal3 localization was not affected by motor concentration (Fig. 3d and Extended Data Fig. 4b). Instead, to represent the effects of Mal3, we assumed different motor/cargo behaviours at microtubule end and lattice, and studied different scenarios for cargo oligomerization.

We first investigated the effect of motor slowdown in the comet region near the microtubule plus-end. Mimicking the hydrolysis state of GTP using GTPγS microtubules, we observed that motor intensity increased and motor speed slowed down (Extended Data Fig. 7a–c). Simulations show that motor slowdown near the microtubule end indeed leads to end accumulation (Fig. 6b), an effect that is due to a traffic jam at the transition from the fast to the slow parts of the lattice (dashed line in Fig. 6b). This type of traffic jam is different from the previously reported formation of 'spikes'[39], which is due to a reduced motor off-rate at the microtubule end.

Inspired by our experimental observations in guest-host and end-tracking assays, by evidence that Tip1 may be able to oligomerize[8,46], and by structural data suggesting interactions between Tip1's CAP-Gly domain and its C-terminal zinc finger domain[13,47], we next considered the effect of lateral interactions between neighbouring motor-bound cargo molecules (Fig. 6c). When we simulated cargo particles as synchronously moving oligomeric cargo trains[48], the effective flux of cargo on the microtubule increased, but in the absence of any end-specific effects, this did not lead to accumulation of cargo at the microtubule end (Fig. 6c). Accumulation of cargo was, however, recovered by introducing motor slowdown at the microtubule end as in Fig. 6b.

Finally, we explored the effect of increased stability of cargo clusters: motor-cargo neighbouring with at least one other motor-cargo was given a higher dwell time compared with non-clustered motors (Fig. 6d). Even a three-fold increase in dwell time was not sufficient to cause end accumulation (Fig. 6d). However, addition of the end-dependent slowing down resulted in pronounced cargo accumulation at the microtubule end (Fig. 6d). Note that stabilization of oligomeric clusters also increases the lattice occupancy away from the microtubule end (Fig. 6d,e). For all scenarios, we also investigated how the accumulation of the cargo depends on the concentration of motors in the model. Only the scenario in which motors slow down at the microtubule end and cargo clusters are stabilized by lateral interactions resulted in non-linear end accumulation of cargo (Fig. 6e,f).

## Discussion

In this study, we systematically dissected the role of multivalent interactions within the MPET network reconstituted in vitro using recombinant Mal3, Tip1 and Tea2 from *S. pombe*. We found that in vitro molecular crowding agents, such as PEG, drove these proteins into spherical droplets that displayed liquid-like properties: they fused with each other over time, wetted the microtubule surface and transferred from one microtubule to another. This behaviour shows similarity to the

previously observed transfer of end-tracking protein clusters from a microtubule end to a solid barrier[8], and might be relevant in vivo for the cortical deposition of polarity markers that are crucial for the physiology of fission yeast such as Tea1, Tea4 and Tea3 in addition to Mal3, Tea2 and Tip1 (refs. [49–53]).

Under crowding conditions Mal3, Tip1 and Tea2 co-existed in the same condensed phase. Although interactions between these proteins were reported previously, it remained unclear which domains of Mal3 were involved[10,11,15]. Here we show that Mal3 IDR1 and IDR2 are responsible for interactions with Tip1 and Tea2 in the absence of microtubules (Fig. 5c and Extended Data Fig. 5a,b). Deletion of these disordered regions impaired formation of Mal3 droplets in crowding conditions (Fig. 5a), in accordance with the idea that disordered regions are the main drivers of LLPS[16,17]. We further found that Mal3 IDR1, in combination with EB HD, is crucial for Mal3's accumulation at growing microtubule plus-ends (Fig. 4 and Extended Data Fig. 6).

Importantly, Mal3-ΔIDR2, for which we also observed severely impaired droplet formation and interaction with Tip1 and Tea2 in crowding conditions, was nevertheless able to form comets at growing microtubule ends (Fig. 4b–d) and recruit Tip1 to these comets (Fig. 5e,g). However, in comets formed by Mal3-ΔIDR2, both Mal3 and Tip1 intensity were reduced, and Mal3 association with the lattice (Fig. 4b,c) and Tea2 transport (Fig. 5e,g) was also reduced. We conclude that robust motor-driven transport and accumulation of Tip1 at microtubule ends depend on both Mal3 IDR1 and IDR2, leading to the suggestion that Mal3 self-interactions responsible for LLPS are also responsible for protein interactions in the network-like structures observed at microtubule ends in cryo-ET (Fig. 2e,f).

The question that then remains is whether the network-like structures observed in cryo-ET in the absence of crowding agents show characteristics of liquid-like droplets and/or whether this is expected to be the case for end-tracking complexes in vivo. Clearly, PEG-driven droplets in the absence of microtubules displayed a characteristic dense internal grain in cryo-ET (Fig. 2b,c) that was not seen in the microtubule end-tracking comets. The comets appeared as loosely structured densities (Fig. 2e,f) which did not extend further than 55 nm from the microtubule lattice (Fig. 2g,h). This even remained the case when Mal3/Tip1/Tea2 together with PEG were added to pre-formed comets (Fig. 2e), when we sometimes observed an additional layer around the comet. Given the estimated dimensions of Mal3 (3 × 6 × 10 nm) (refs. [54,55]), Tea2 (4 × 4 × 7 nm) and Tip1 (predicted 40-nm-long coiled coil), it is technically possible that all the molecules inside the comet are directly interacting with microtubule surface. On the other hand, it is also possible that the loose network represents a liquid-like structure where multiple dynamic, weak interactions between its components facilitate the observed non-stoichiometric accumulation of plus-end trackers and allow them to behave as a protein cluster[8]. In vivo, where crowding effects could be different from the conditions of our cryo-ET experiments, these clusters may again appear as dense droplets as we observed in the presence of PEG. We must also note that preserving droplets at the ends of microtubules during our cryo-ET sample preparation may be technically challenging, potentially limiting our ability to visualize these structures.

In conclusion, our study suggests that microtubule ends may act as platforms where multivalent interactions condense microtubule-associated proteins into large complexes. Our observations are paralleled by observations in three other biological systems: formation of Kar9 nanodroplets at the ends of specialized microtubules in budding yeast[56], formation of droplets by human EB3 and CLIP-170 at microtubule ends[57], and condensation of EB1 affecting chromosome segregation during mitosis[58]. LLPS at microtubule ends is thus emerging as a general organizing principle that may explain how different end-tracking proteins may (simultaneously) associate with microtubule ends to perform their wide range of biological functions.

## Online content

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

## Methods

### Protein expression, purification and labelling

Full-length *S. pombe* Mal3 and all of its derivatives (that is, truncations and superfolder-GFP fusions) were expressed with an N-terminal His8 tag followed by a 3C protease site from a pBADTOPO derived plasmid in *Escherichia coli* ER2566 cells (New England Biolabs, *fhuA2 lacZ::T7 gene1 [lon] ompT gal sulA11 R(mcr73::miniTn10–Tet$^S$)2 [dcm] R(zgb-210::Tn10–Tet$^S$) endA1 Δ(mcrCmrr)114::IS10)*). Mal3(-truncations) were covalently linked to superfolder-GFP by a flexible

ASTGILGAPSGGGATAGAGGAGGPAGLINPGGSTSSRAAEIWPAS 'happy linker' sequence.

Cells were grown at 37 °C in baffled flasks on LB supplemented with 100 μg ml$^{-1}$ ampicillin, expression was induced at an OD$_{600}$ of 0.6, and cells were collected after 3 h (8 min 4,500 rpm, JLA8.1000 rotor). After washing the cells in PBS, they were lysed using a French Press (Constant Systems) at 20 kpsi, 4 °C, and unbroken cells, debris and aggregates were pelleted in a Ti45 rotor (30 min, 40,000 rpm, 4 °C). The lysate was applied to 2 ml Talon Superflow resin (Clontech) pre-equilibrated with buffer A (20 mM Tris–HCl pH 7.5, 200 mM NaCl and 5% (w/v) glycerol), and incubated for 1 h while rotating at 4 °C. Subsequently, the resin was washed with 50 ml of buffer A supplemented with 0.1% Tween20 and 50 ml of buffer A supplemented with additional 500 mM NaCl, and finally Mal3 was eluted in 10 ml of buffer A supplemented with 1 mM β-mercaptoethanol and homemade 3C protease. Proteins were concentrated using a Vivaspin centrifugal concentrator (10 kDa cut-off) and further purified by size exclusion chromatography (SEC) on a Superdex 200 Increase 10/300 column pre-equilibrated with buffer B (20 mM Tris–HCl, 100 mM NaCl and 5% (w/v) glycerol).

Mal3 was labelled by dialysing ~1 mg of protein into buffer C (80 mM PIPES pH 6.8, 1 mM MgCl2 1 mM ethylene glycol tetraacetic acid (EGTA) and 100 mM NaCl) and incubating for 1 h at room temperature with 140 μM Alexa Fluor 488 TFP ester or Alexa Fluor 647 TFP ester (Thermo Fisher). After quenching the reaction with excess Tris–HCl, the free label was removed by SEC on a Superdex 200 Increase 10/300 column pre-equilibrated with buffer B.

Full-length *S. pombe* Tea2 was expressed with an N-terminal Z-tag followed by a TEV protease recognition site, and purified essentially as described[9], but with the following modifications: after washing of the Talon resin with 15 mM imidazole in buffer D (50 mM KPi pH 8.0, 400 mM NaCl, 2 mM MgCl$_2$, 0.2 mM MgATP and 0.05 mM TCEP), Tea2 was eluted in buffer D supplemented with homemade 3C protease by taking advantage of crossreactivity with the TEV recognition site. Following concentration using a Vivaspin centrifugal concentrator (10 kDa cut-off), Tea2 was labelled with 138 μM Alexa Fluor 647 NHS ester (Thermo Fisher) by incubating 30 min at room temperature. After quenching the reaction with excess Tris–HCl, the free label was removed by SEC on a Superdex 200 Increase 10/300 column pre-equilibrated with buffer D. Unlabelled Tea2 was applied to the SEC column directly after concentrating.

### Flow cell preparation

Coverslips and glass slides were cleaned using base Piranha (NH$_4$OH:H$_2$O$_2$ in 3:1 at 75 °C) for 10 min and sonicated in MilliQ water for 5 min. Flow cells were prepared by sandwiching two strips of parafilms between the glass slide and the coverslip. The strips were placed about 3–5 mm apart approximately from each other. The flow cell was then placed on top of a hot plate, kept at 120 °C, to let the parafilm melt and seal the glass slid with the coverslip.

### Microtubule biochemistry

**GMPCPP-stabilized microtubule seeds.** Microtubule seeds were prepared by two cycles of polymerization with GMPCPP in MRB80 buffer (80 mM PIPES, 4 mM MgCl$_2$ and 1 mM EGTA, pH 6.8). First 20 μM tubulin (25% HyLite 647, 10% biotinylated and 65% unlabelled) was polymerized in the presence of 1 mM GMPCPP (NU-405 Jena BioScience) at 37 °C

for 30 min. The mix was centrifuged for 5 min at 200,000 *g* with an air-driven ultracentrifuge, airfuge (Beckman Coulter), and the pellet was resuspended in MRB80 (80% of the initial volume) and kept on ice for 20 min for depolymerization. For the second polymerization step, again 1 mM of GMPCPP was added to the mix and the mix was incubated at 37 °C for another 30 min. After 30 min of incubation the mix was ultracentrifuged using an airfuge at 200,000 *g* and the pellet was resuspended in 50 μl MRB80 with 10% glycerol. The seeds thus prepared were aliquoted, flash frozen in liquid nitrogen and stored at −80 °C.

**End-tracking reconstitution assays.** To functionalize the glass surface, the channels in the flow cells were first filled with 0.2 mg ml$^{-1}$ PLL( 20)-g[3.1]-PEG(2)/-PEG(3.4)-biotin(17.5%) (SUSOS AG) then 0.1 mg ml$^{-1}$ neutravidin and finally with κ-casein (Sigma). Ten minute incubation at room temperature was maintained before the subsequent steps. The channels were then washed with MRB80 and incubated with biotinylated seeds for 5 min. After 5 min, the reaction mix was added to the channels. The channels were sealed with VALAP before starting the observations on the microscope to avoid evaporation.

To reconstitute plus-end-tracking assays with full-length Mal3 and Mal3 mutants, the reaction mix contained 200 nM Mal3/Mal3 construct, 20 nM Tea2 and 150 nM Tip1 in MRB80 buffer containing 14.5 μM tubulin, 0.5 μM rhodamine tubulin, 50 mM KCl, 0.5 mg ml$^{-1}$ κ-casein, 0.4 mg ml$^{-1}$ glucose oxidase, 50 mM catalase, 0.1% methylcellulose, 1 mM GTP and 2 mM ATP.

### Phase separation assays

**Condensates on dynamic microtubules.** The assay was performed in two steps. In the first step a dynamic microtubule assay was set up in a flow cell and in the second step condensates were added. To set up a dynamic microtubule assay, a reaction mix with 14.5 μM tubulin, 0.5 μM rhodamine tubulin, 50 mM KCl, 0.5 mg ml$^{-1}$ κ-casein, 0.4 mg ml$^{-1}$ glucose oxidase, 50 mM catalase, 0.1% methylcellulose, 1 mM GTP and 2 mM ATP in MRB80. The flow cell was then left for incubation at 37 °C for 15 min. After 15 min the tubulin was washed off using MRB80 (pre-warmed at 37 °C) and condensates were added to the flow cell immediately to the flow cell.

The condensates were prepared by incubating 200 nM Mal3, 20 nM Tea2 and 150 nM Tip1 in MRB80 buffer containing 50 mM KCl, 0.5 mg ml$^{-1}$ κ-casein, 0.4 mg ml$^{-1}$ glucose oxidase, 50 mM catalase, 0.1% methylcellulose, 1 mM GTP and 2 mM ATP on ice for 1 h with 5% PEG-35k.

**Scaffold-client experiments.** Coverslips were cleaned as described above. Glass slides were cleaned in a 250 ml beaker with a custom-made Teflon rack by repeated (2×) sonication and washing steps as follows 1% Hellmanex (10 min), MilliQ water (5 min), 70% ethanol (10 min), MilliQ (5 min) and stored in the beaker with MilliQ covered by parafilm. Before use, slides were rinsed with MilliQ and dried with N$_2$.

Flow cells were prepared by cutting six channels into a piece of parafilm with a razor blade. The parafilm was sandwiched between the clean glass slide and the cover glass and heated on a piece of aluminium foil on top of a 120 °C hot plate until the parafilm melted and cover glass was gently pressed with tweezers to assure that channels were sealed off well. The parafilm overhangs were removed with the blade while the glass was still hot. After cooling to room temperature, the channels were incubated for 10 min with 0.2 mg ml$^{-1}$ PLL(20)-g[3.1]-PEG(2) (SUSO AG), rinsed and incubated for 10 min with 0.5 mg ml$^{-1}$ κ-casein (Sigma), all solutions were MTB80 buffer.

The scaffold and client condensates were prepared on ice by first eluting all proteins into MRB80 buffer containing 250 mM KCl, and then further diluting them into the reaction mixture, at a final composition of 1× MRB80, 50 mM KCl, 10% PEG6k and freshly thawed 2 mM ATP, 1 mM GTP, 2 mM dithiothreitol and 0.5 mM β-mercaptoethanol. Solutions were clarified for 5 min at 200,000 *g* using an airfuge and kept

on ice for 15 more minutes before being transferred into flow cells. Imaging occurred approximately 30 min after mixing. Mal3 and Tip1 host condensates were prepared with 200 nM FL Mal3 or 215 nM Tip1, 2 nM FL Mal3:mCherry and 2 nM of each of the constructs (Fig. 3a). Tea2 scaffold condensates were prepared from 200 mM Tea2-Alexa647 and 2 nM of each of the constructs. Experiments for each scaffold protein were conducted in parallel. Image acquisition was performed using spinning disc confocal microscopy (CSU-W1, Yokogawa; Ilas2, Roper Scientific) with the scanning slide module in the Ilas2 software.

## Cryo-electron tomography
To study PEG-driven droplets, a solution containing 200 nM Mal3, 150 nM Tip1 and 80 nM Tea2 was incubated with 10% of PEG-8k in MRB80; 4 µl of this solution was mixed with 5 nm gold nanoparticles (OD50, final dilution 1:20) and added to freshly glow-discharged copper grids with R2/2 Quantifoil film. The grid was blotted from the back side for 4–6 s in a Leica EM GP plunger and immediately plunge-frozen in liquid ethane.

To reconstitute comet formation, we used copper mesh grids with holey SiO film (SPI Supplies), coated with 5 nm gold on one side. The grids were treated with oxygen plasma for 2 min and immediately submerged in Plus-One Repel Silane solution (GE Life Sciences) for 3 min, then washed in ethanol and dried. A silanized grid was incubated in a drop of anti-DIG IgG (0.2 µM, Roche), washed with MRB80, incubated in a drop of 1% Pluronic F-127 and washed again with MRB80. The passivated grid was then taken into the chamber of the Leica EM GP2 plunger equilibrated at 95% relative humidity and 26 °C. Inside the chamber, GMPCPP-stabilized, DIG-labelled microtubule seeds were added to the grid for 1 min followed by a wash with MRB80 supplemented by 0.5 mg ml$^{-1}$ κ-casein and finally a 4 µl drop of a solution containing 200 nM Mal3, 150 nM Tip1 and 80 nM Tea2 in MRB80 supplemented with 25 µM tubulin, 0.01% Tween20, 2 mM ATP, 1 mM GTP and 1 mM dithiothreitol. The microtubules were allowed to grow for 7 min, after which 5 nm gold nanoparticles were added (OD50, final dilution 1:20), the grid was blotted from the back side for 3–4 s and immediately plunge-frozen in liquid ethane. All grids were stored in closed boxes in liquid nitrogen until further use.

Tilt series were recorded on a JEM3200FSC microscope (JEOL) equipped with a K2 Summit direct electron detector (Gatan) and an in-column energy filter operated in zero-loss imaging mode with a 30 eV slit width. Images were recorded at 300 kV with a nominal magnification of 10,000×, resulting in the pixel size of 3.668 Å at the specimen level. Automated image acquisition was performed using SerialEM 3.8.5. software[60] with a custom-written script, recording bidirectional tilt series ranging from 0° to ±60° with tilt increment of 2°; a total dose of 100 e$^-$ Å$^{-2}$ and the target defocus set to −4 µm. Individual frames were aligned using MotionCor2 (ref. [61]), and then split into odd and even frame stacks. Tilt-series alignment and tomographic reconstructions were performed with the IMOD software package using gold beads as fiducial markers[62]. Final tomographic volumes were binned two-fold and subsequently denoised using the cryoCARE procedure[32]. For this, 3D reconstruction was performed on aligned sets of odd and even frame stacks with identical IMOD parameters. The full even and odd tomograms obtained in this way were then split into subvolumes for network training, and eventually full volumes were denoised. The images shown in Fig. 5 were obtained from a voxel-wise average of odd and even denoised tomograms. Automated segmentation of binned and denoised cryo-tomograms was performed using the *tomoseg* module of EMAN2 v.2.2 (ref. [63]) and visualized using UCSF Chimera[64].

## TIRF microscopy
Imaging was performed using an inverted Nikon Eclipse Ti-E microscope with perfect focus system, an oil immersion objective (Nikon Plan Apo λ 100× NA 1.45), using two EMCCD cameras (Photometrics Evolve 512), which are mounted on a spinning disc unit (CSU-W1,

Yokogawa). TIRF illumination was generated with the FRAP/TIRF system Ilas2 (Roper Scientific). A custom-made objective heater was used for temperature control of the samples. The imaging software used was Metamorph 7.8.8.0 with system specific routines (Ilas2) for streaming, time lapse and scanning slide acquisition.

## Stochastic simulations
Stochastic simulations were performed using Gillespie's algorithm[65] on the TU Delft Applied Science in-house linux cluster using an implementation in C++. The different implementations of the model were all simulated, tested and prepared independently. Model parameters were chosen as much as possible in agreement with experimental conditions and corresponded to a low-density regime (LD phase) in terms of the TASEP/LK model on growing microtubules[42].

System size was 1,000 lattice sites, each corresponding to the size of one tubulin heterodimer (8.4 nm). Simulations were equilibrated for $10^5$ s before $10^4$ motor and cargo distributions were recorded in time intervals corresponding to the time it takes for one motor to traverse the system (~50 s). Equilibration times were particularly critical for cargo clustering conditions, since the motor distributions generically deviate from their classical equilibrium owing to the aggregation and fragmentation kinetics, as seen in similar systems[48]. Data analysis and plotting was performed using custom programs and scripts written in C++ and Python (Matplotlib). Details regarding all model parameters and corresponding experimental values can be found in Supplementary Table 2.

## Data analysis
**Preparation of density profiles.** Kymographs were extracted from background-subtracted TIRF microscopy data in a semiautomated way using ImageJ (50 pixel rolling ball radius). Image projections were used to identify dynamic microtubules in movies (function Z Project with option standard deviation) and positional data were stored in the form of linear regions (thickness 9 pixels) using the ImageJ ROI manager. Saved regions of interest were used to automatically generate kymographs.

Subsequently a MATLAB (R2018b) script was used to analyse kymographs (dual colour where necessary) in a semi-automated way. The script allows to manually mark regions of growing microtubules with comets as polygons (typically triangular), generates a mask thereof, and extracts the corresponding intensity profiles from the underlying images. The intensity profiles are saved per experimental condition for further processing.

In a separate step, the intensity profiles were sorted into sets by length using a binning of (±0.64 µm). The set of profiles was aligned by finding alignments which minimize the s.d. of the sum of differences between a randomly chosen first intensity profile and every other profile in a set. Averages of the aligned sets of data are shown in Extended Data Fig. 4c,d. Regions of end and lattice intensities were defined manually. For Fig. 3e, individual intensities at 100 nM Tea2 (for all microtubule lengths) were divided by the average of all intensities at 20 nM Tea2.

**Analysis of scaffold-client experiments.** Analysis of scaffold-client experiments was performed using a custom script written in MATLAB (R2018b) including the image processing toolbox. Fluorescence microscopy images of scaffold and client condensates were loaded after rolling ball (50 pixel) background subtraction using ImageJ. Condensates were identified in the mCherry or Alexa-647 fluorescence channel ('tag'). The positional information was used to quantitatively evaluate co-localization of Mal3:GFP (client molecules; Fig. 5b and Extended Data Fig. 5a,b) and the tag (Extended Data Fig. 5c).

The procedure consisted of converting the mCherry/Alexa-647 image to a binary image that can be used as an image mask (im2bw function with manually optimized threshold levels ~0.05). The image

mask was then used to detect condensates and evaluate their positions, major and minor axes lengths, and the mean intensity, using the regionprops function for centroid regions.

**Analysis of Mal3/Tea2/Tip1:GFP velocities.** Single-molecule traces of Mal3/Tea2/Tip1:GFP complexes were recorded at concentrations of 200 nM Mal3, 1 nM Tea2 and 150 nM Tip1:GFP under MPET conditions. A total number of $N = 148$ Tip1:GFP traces remained after automated detection in seven kymographs using KymoButler[66], and manual exclusion of obscure traces (crossings, merging or tracks that reach the microtubule end). We calculated a median of 0.23 µm s$^{-1}$ and a standard error of the mean of 0.06 µm s$^{-1}$. The velocity of Mal3/Tea2/Tip:GFP clusters in the presence of PEG35k (Fig. 1h) was assessed after transfer events between microtubule ends and surrounding microtubules. We manually measured $N = 47$ traces with a median velocity of 0.12 µm s$^{-1}$ and a standard error of the mean of 0.018 µm s$^{-1}$ (Extended Data Fig. 7d).

### Statistics and reproducibility
Data reported are from at least three independent repeats for each experiment. Detailed information on reproducibility for individual experiments is available from respective figure legends. *P* values are reported as a result of the Mann–Whitney test. No statistical method was used to pre-determine sample size. No data were excluded from the analyses; the experiments were not randomized; the investigators were not blinded to allocation during experiments and outcome assessment.

### Reporting summary
Further information on research design is available in the Nature Portfolio Reporting Summary linked to this article.

### Data availability
Tomography data shown in Fig. 2 are available from Electron Microscopy Data Bank (EMDB) using the following accession codes: microtubule plus-end in presence of Tip1, Tea2 and Mal3 (EMD-14110), microtubule minus-end in presence of Tip1, Tea2 and Mal3 (EMD-14111), microtubule plus-end in presence of Mal3 (EMD-1408), microtubule minus-end in presence of Mal3 (EMD-14109), microtubule plus-end in absence of additional proteins (EMD-14106), microtubule minus-end in absence of additional proteins (EMD-14107), Tip1, Tea2 and Mal3 in presence of PEG without microtubules or tubulin (EMD-14112) and Tip1, Tea2 and Mal3 in presence of both PEG and microtubules (EMD-14182). Source data are provided with this paper. All other data are available upon request.

### Code availability
Python scripts used for splitting of movie frames, reconstruction of even and odd tomographic volumes, training data generation, model training and denoising are available at https://github.com/NemoAndrea/cryoCARE-hpc04. The simulation code is available at https://github.com/luiree/TipPhase.

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

### Acknowledgements

We are grateful to all group members of the Dogterom as well as Akhmanova (Utrecht University) labs for many discussions on microtubule end tracking proteins during ERC Synergy meetings. Initial droplet assays were performed during the Physiology Course 2017 at the Marine Biological Laboratory in Woods Hole. This work was supported by the following grants awarded to M.D.: FOM programme number 110 from the Netherlands Organisation for Scientific Research (L.R.), European Research Council Synergy grant 609822 (V.A.V.) and Sinergia grant 160728 from the Swiss National Science Foundation (E.O.v.d.S. and R.M.).

### Author contributions
All authors planned experiments, analysed data and discussed results. E.O.v.d.S. purified the proteins. R.M. performed TIRF experiments including Tea2 titration experiments. R.M. and L.R. performed droplet assays, following the initial observation by M.R.K. L.R. performed guest-host assays and theoretical modelling and analysed data of Tea2 titration experiments. V.A.V. performed electron microscopy and 3D reconstruction with help of A.J.J. W.H.E. coated grids with gold and contributed tilt-series acquisition scripts. N.A. contributed cryo-CARE automation scripts. V.A.V., R.M., L.R. and M.D. wrote the paper.

### Competing interests
The authors declare no competing interests.

### Additional information
**Extended data** is available for this paper at https://doi.org/10.1038/s41556-022-01037-0.

**Correspondence and requests for materials** should be addressed to Marileen Dogterom.

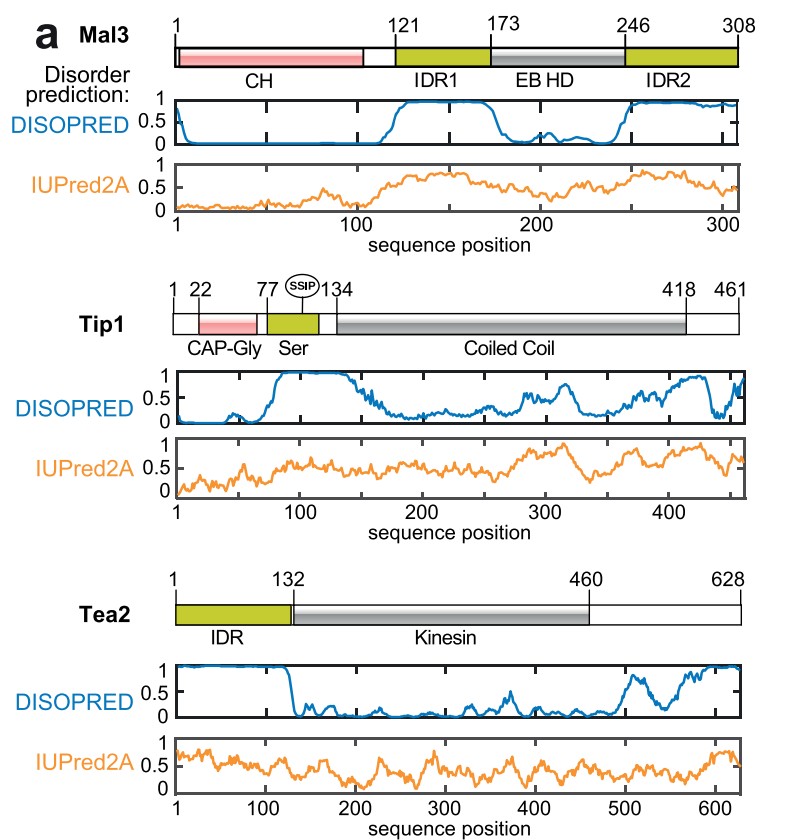

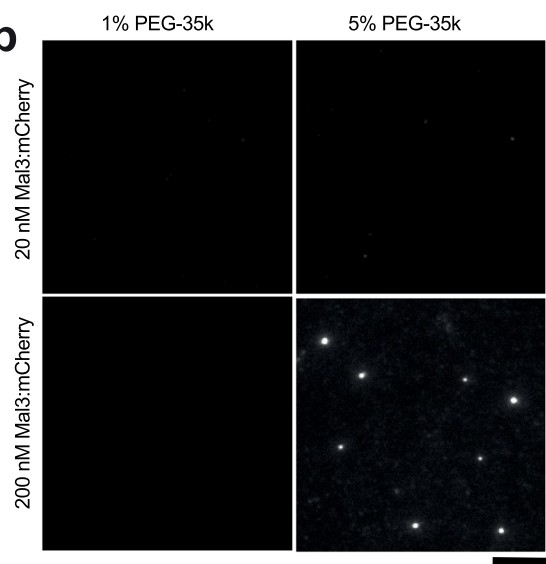

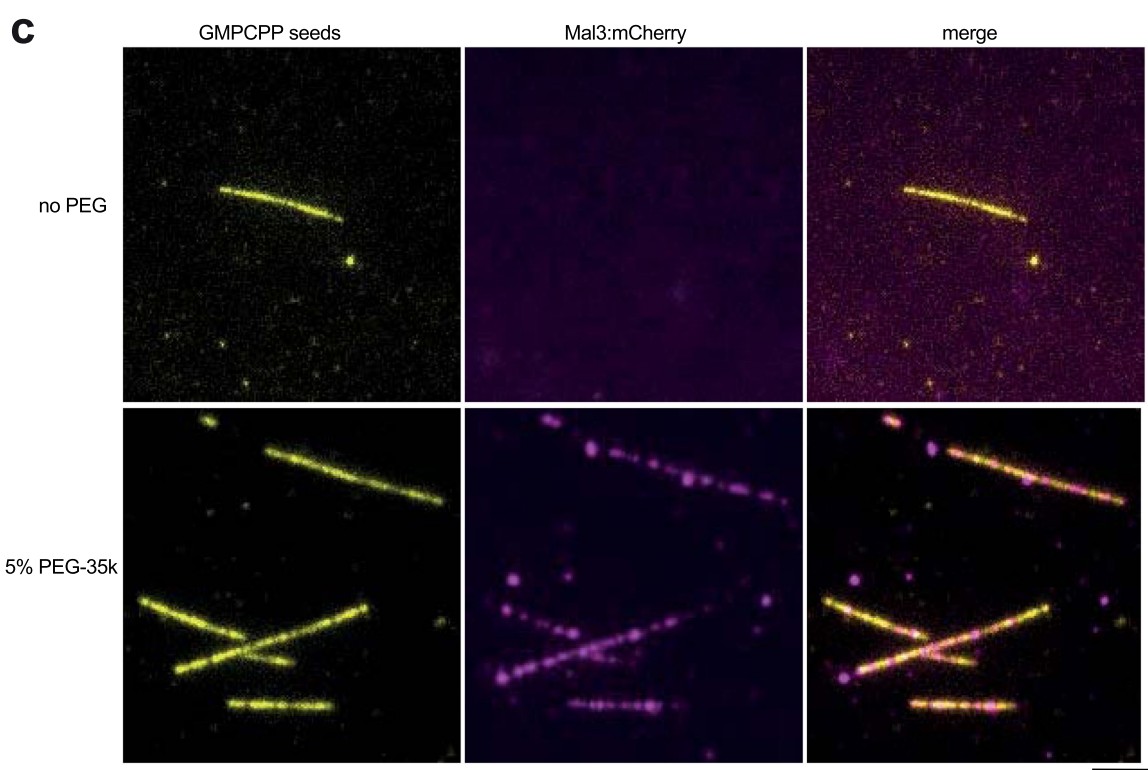

**Extended Data Fig. 1 | (a) Protein regions of Mal3, Tip1 and Tea2 together with disorder prediction (DISOPRED, blue, and IUPred2A, orange).** In particular, the serine-rich part of Tip1 and tails of Tea2 are predicted differently in DISOPRED3 and IUPred2A. **(b)** Titration of PEG35k percentage and Mal3 concentration required to achieve droplet formation. **(c)** Mal3:mCherry interaction with fluorescently labelled GMPCPP seeds in the absence of PEG (top row) or in the presence of 5% PEG-35k. Experiments were repeated twice, representative images from one repeat are shown. Scale bars: 5 μm.

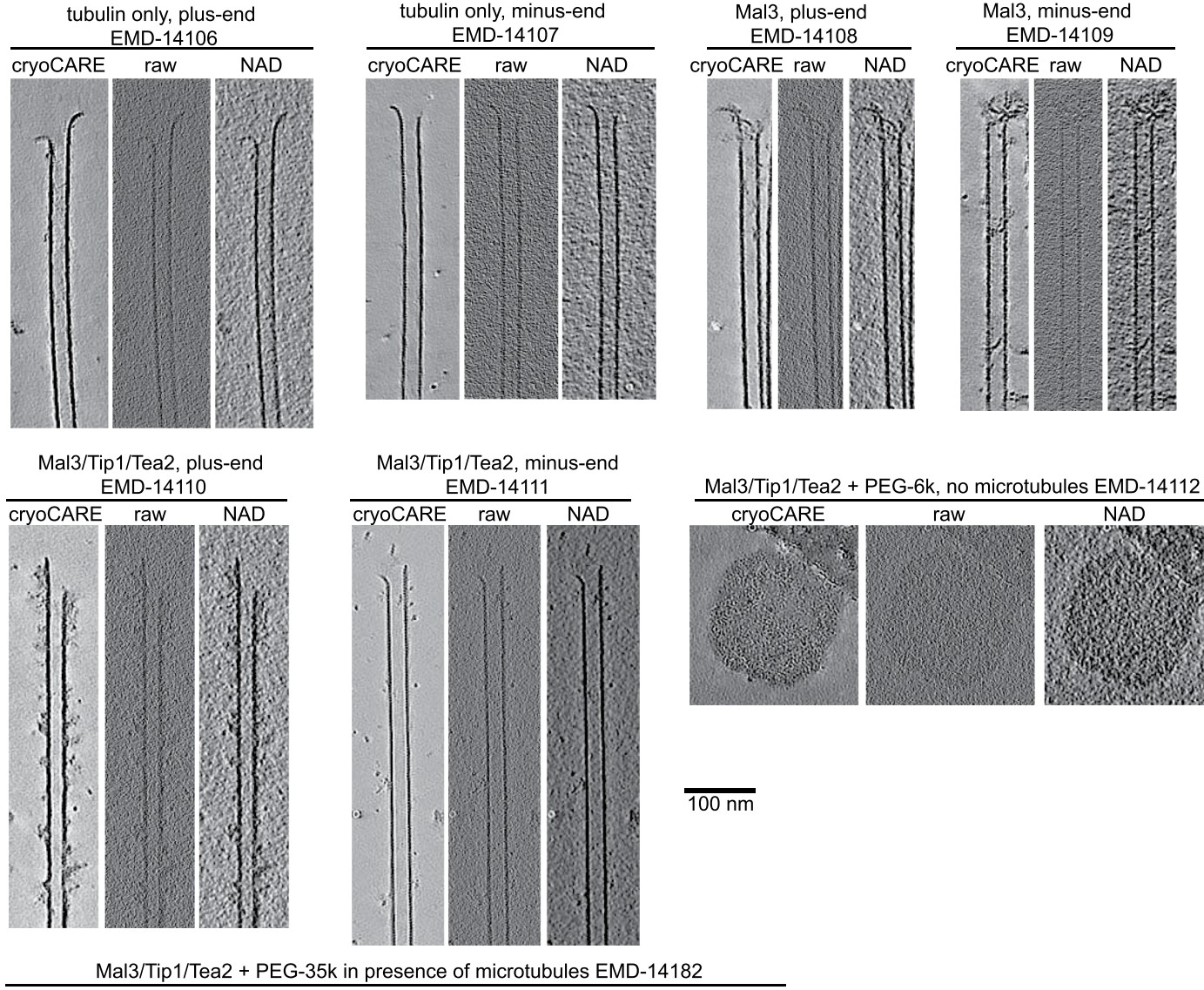

**Extended Data Fig. 2 | Each panel shows a slice from one of the tomograms in Fig. 2 b,e** processed in the following way: (left) cryoCARE-denoised (see Materials and Methods for details), (center) back-projected volume generated using IMOD without further processing, (right) the same volume processed using nonlinear anisotropic diffusion algorithm in IMOD (k = 0.5, 50 iterations). Unprocessed tomograms are available from EMDB using the accession numbers provided for each condition. Experiments were repeated three times, representative images from one repeat are shown.

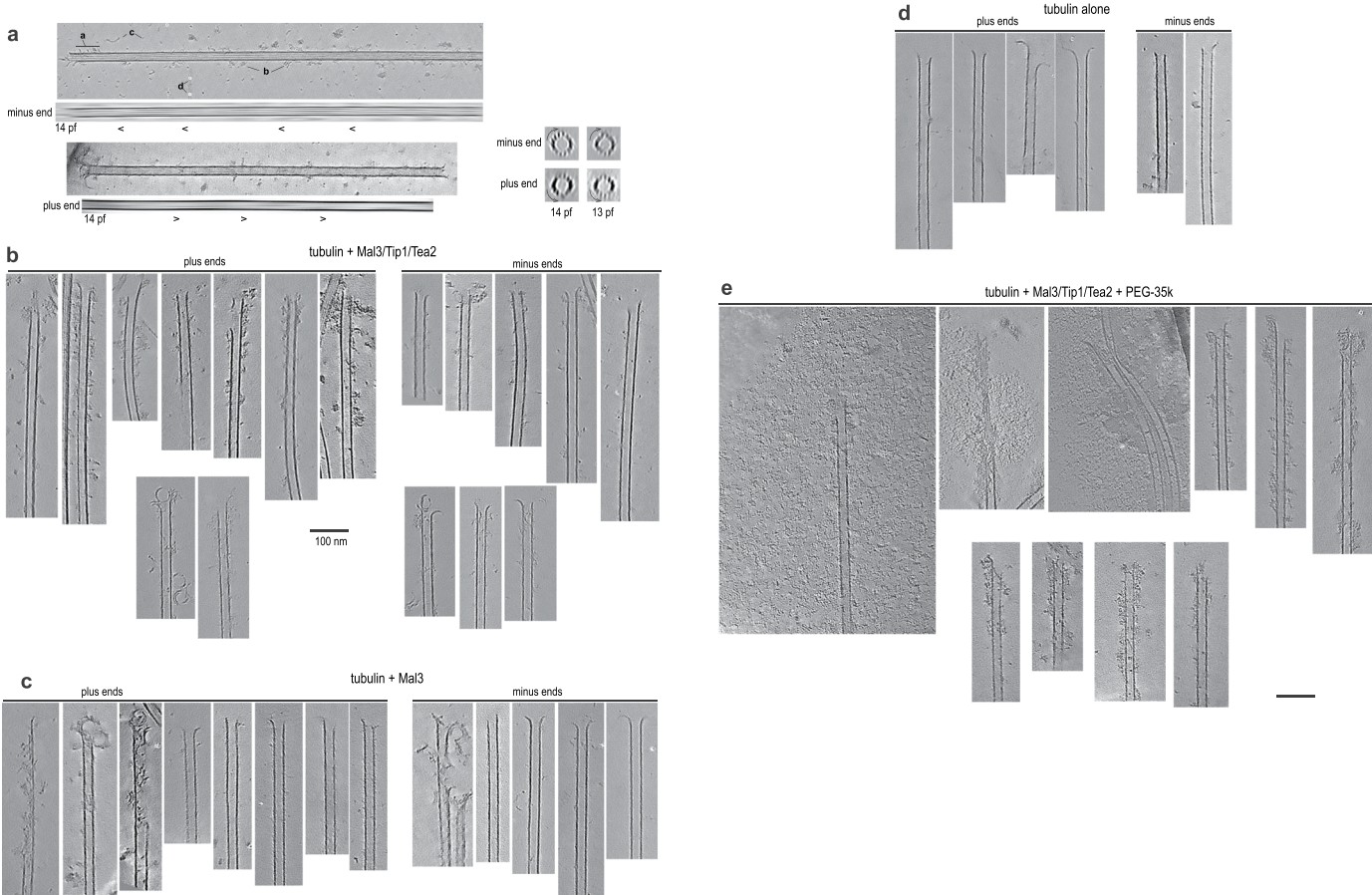

**Extended Data Fig. 3 | (a) Summed tomogram slices containing microtubules (top) and Fourier-filtered subsets containing microtubule moiré pattern (bottom).** Two examples are shown: a 14-protofilament microtubule with the minus end pointing up, and a 14protofilament microtubule with the plus-end pointing up. a – comet at the minus-end, b – Mal3/Tip1/Tea2 oligomers bound to microtubule lattice, c – soluble tubulin oligomers, d – gold particles with erased densities. (b) Examples of plus- and minus-ends of microtubules grown in presence of Mal3, Tip1 and Tea2. Two examples of ends with unclear polarity are shown at the bottom. (c) Examples of plus- and minus-ends of microtubules grown in presence of Mal3. (d) Examples of plus- and minus-ends of microtubules grown in the absence of additional proteins. Scale bars: 100 nm. Experiments were repeated three times, representative images from one repeat are shown.

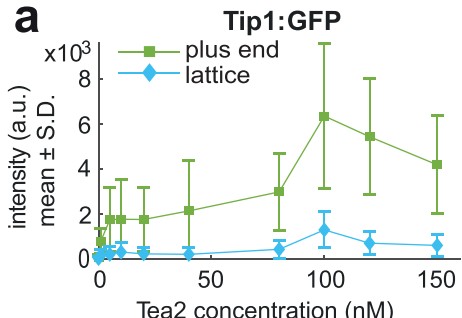

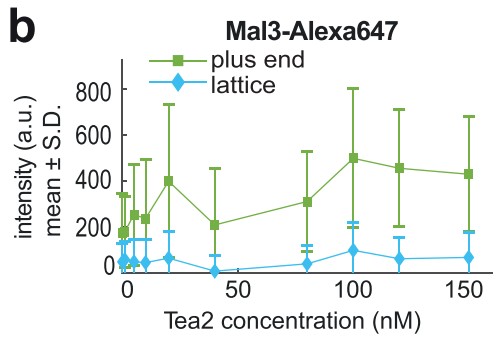

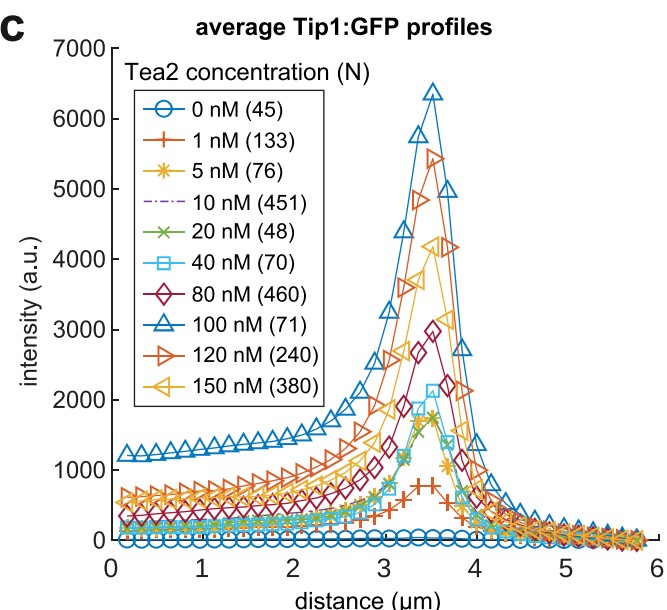

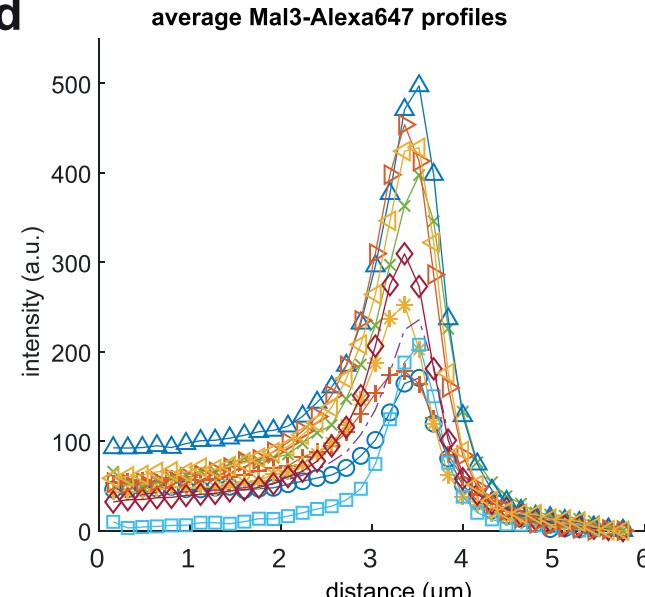

**Extended Data Fig. 4 | (a) Tip1:GFP intensity on the microtubule lattice as well as on the microtubule tip depends non-linearly on the concentration of Tea2.** (b) The intensity of Mal3-Alexa647 is less effected by the Tea2 concentration. Each data point in panels a and b was obtained by averaging tip and lattice intensities at the corresponding concentration, as shown in c and d, respectively. The number of observed intensity profiles obtained from one experiment per condition is indicated in the legend of panel c (the same number for Tip1:GFP and Mal3-647 and for 'tip' and 'lattice', but different at different Tea2 concentrations as indicated). Data are presented as mean ± S.D. (c,d) Intensity profiles of Tip1:GFP (c) and Mal3-647 (d) in the Tea2 titration experiments. Intensity profiles were extracted from TIRF microscopy images of dynamic microtubules.

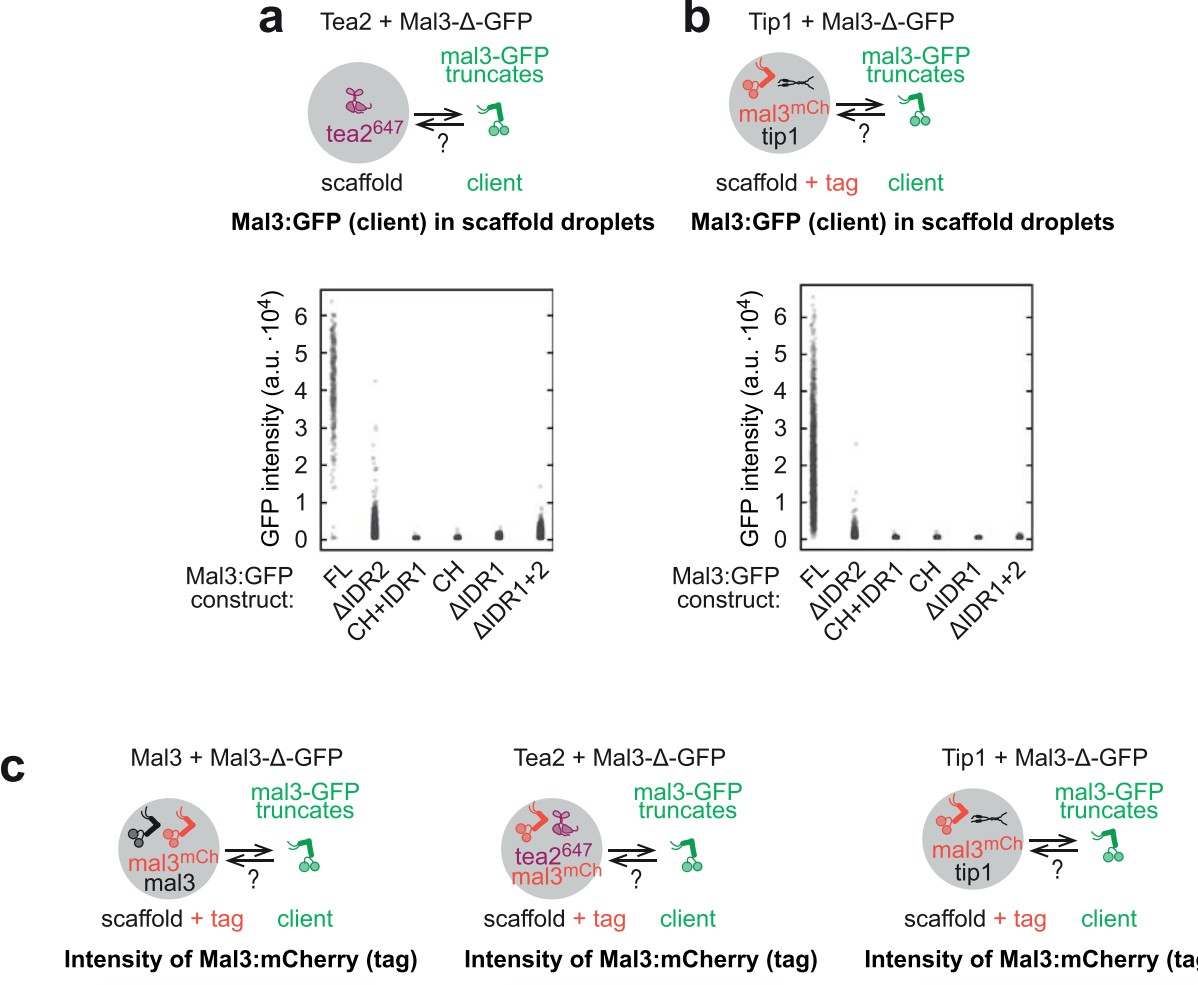

**Extended Data Fig. 5 | (a) Droplets of Tea2-647 (scaffold) were allowed to recruit Mal3:GFP constructs (2 nM, client) in the presence of PEG-6k.** The graph shows distribution of Mal3:GFP construct intensity in the scaffold droplets (number of droplets n = 358, 4178, 5220, 6023, 5317, 6065; from left to right). (b) Droplets of unlabelled Tip1 (scaffold) tagged with FL-Mal3:mCherry (2 nM, tag) were allowed to recruit Mal3:GFP constructs (2 nM, client) in presence of PEG-6k. The graph shows distribution of Mal3:GFP construct intensity in the scaffold droplets (number of droplets n = 1645, 1400, 1248, 831, 908, 1267; from left to right). (c) Intensity of Mal3:mCherry tag in experiments presented in Fig. 4c (Mal3:GFP constructs recruited to unlabelled Mal3 droplets, number of droplets n = 445, 436, 432, 347, 369, 452), and panels a and b. Data was collected on three different days, one experiment per condition.

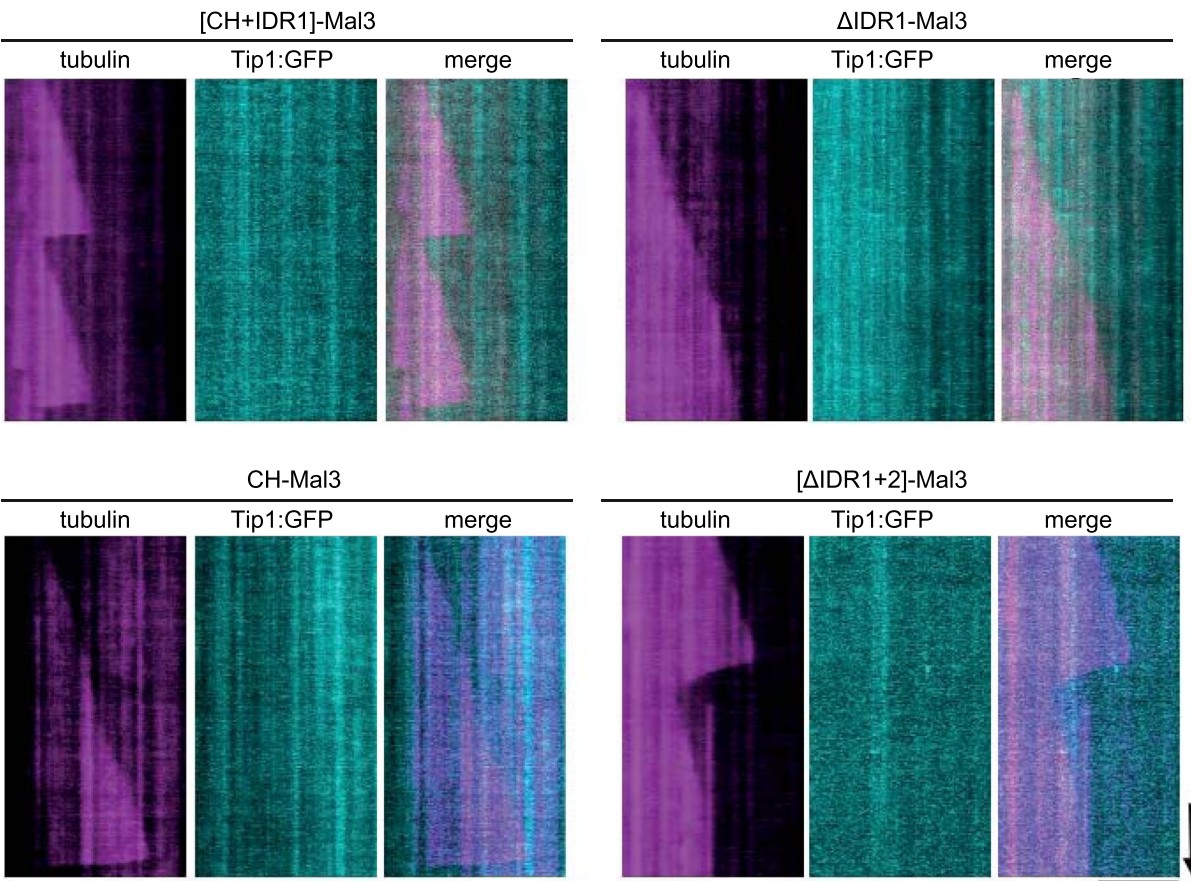

**Extended Data Fig. 6 | Mal3 constructs lacking the disordered region IDR1 neither show motor transport at the lattice nor Tip:GFP accumulation at the plus end.** Scale bars: 5 μm (horizontal) and 60 s (vertical).

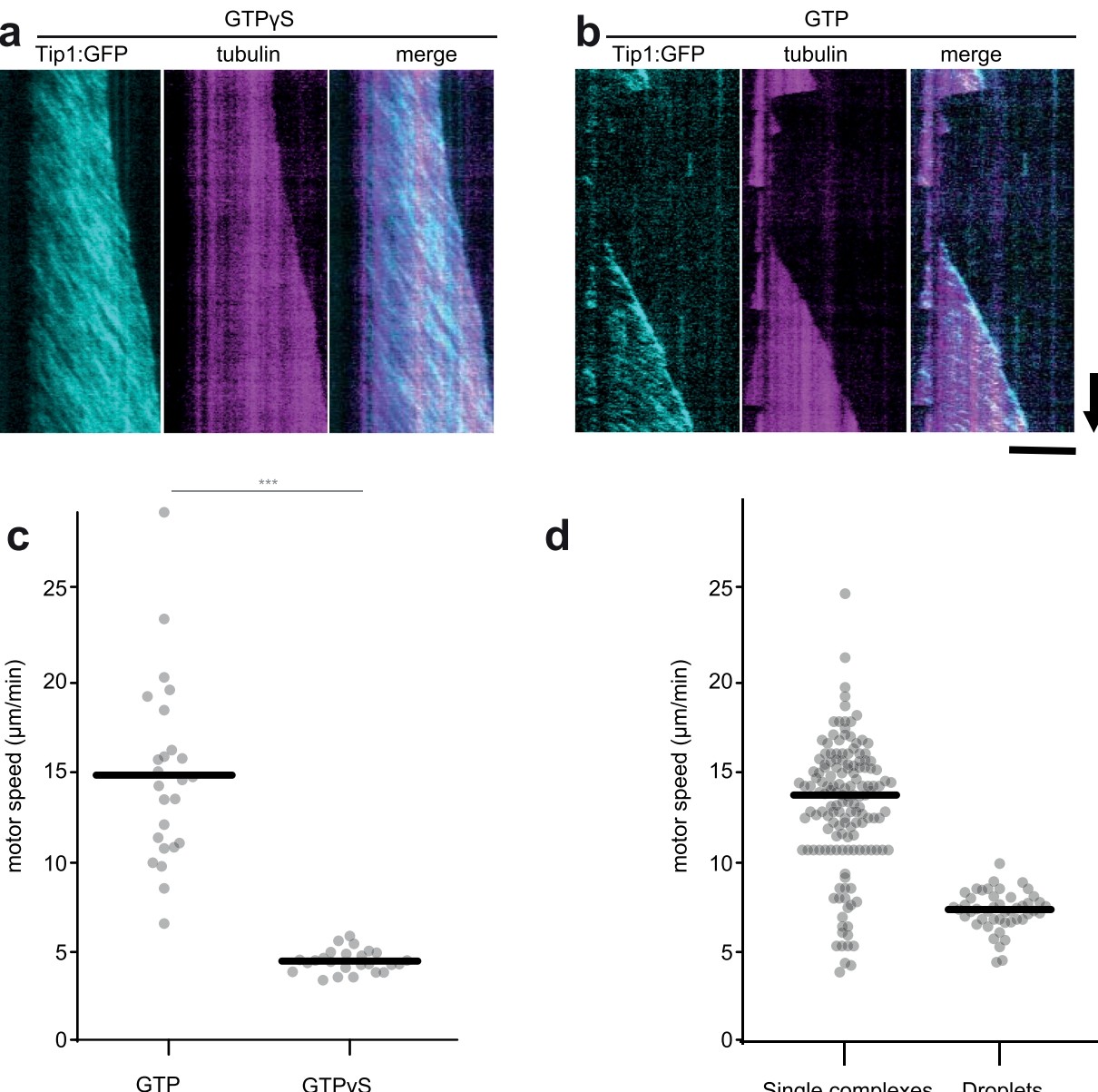

**Extended Data Fig. 7 | (a and b) MPET reconstituted in the presence of 1 mM GTPγS (a) or 1 mM GTP (b).** Scale bars: 5 µm (horizontal) and 60 s (vertical). (c) Speed of motor clusters on the microtubule lattice, estimated from the kymographys in the presence of GTP and GTPγS. Each data point represents the average speed on one microtubule estimated using the imageJ plugin orientationJ. The number of microtubules analyzed for each condition was 25, pooled from two independent experiments. (d) A comparison between the velocities of Mal3/Tea2/Tip1:GFP complexes (200 nM Mal3, 1 nM Tea2, 150 nM Tip1; n = 148 traces from 7 different microtubules in one experiment) and Mal3/Tea2/Tip1:GFP droplets (n = 47 droplet traces from one experiment) in the presence of crowding agents (cf. Figure 1h) shows an approximate 50% reduction in plus-end directed velocity (see Materials and Methods for details).

# Reporting Summary

## Statistics

For all statistical analyses, confirm that the following items are present in the figure legend, table legend, main text, or Methods section.

| n/a | Confirmed | |
|---|---|---|
| ☐ | ☒ | The exact sample size (*n*) for each experimental group/condition, given as a discrete number and unit of measurement |
| ☐ | ☒ | A statement on whether measurements were taken from distinct samples or whether the same sample was measured repeatedly |
| ☐ | ☒ | The statistical test(s) used AND whether they are one- or two-sided<br>*Only common tests should be described solely by name; describe more complex techniques in the Methods section.* |
| ☒ | ☐ | A description of all covariates tested |
| ☒ | ☐ | A description of any assumptions or corrections, such as tests of normality and adjustment for multiple comparisons |
| ☐ | ☒ | A full description of the statistical parameters including central tendency (e.g. means) or other basic estimates (e.g. regression coefficient) AND variation (e.g. standard deviation) or associated estimates of uncertainty (e.g. confidence intervals) |
| ☐ | ☒ | For null hypothesis testing, the test statistic (e.g. *F*, *t*, *r*) with confidence intervals, effect sizes, degrees of freedom and *P* value noted<br>*Give P values as exact values whenever suitable.* |
| ☒ | ☐ | For Bayesian analysis, information on the choice of priors and Markov chain Monte Carlo settings |
| ☒ | ☐ | For hierarchical and complex designs, identification of the appropriate level for tests and full reporting of outcomes |
| ☒ | ☐ | Estimates of effect sizes (e.g. Cohen's *d*, Pearson's *r*), indicating how they were calculated |

*Our web collection on statistics for biologists contains articles on many of the points above.*

## Software and code

Policy information about availability of computer code

| Data collection | Fluorescence data were recorded using Metamorph 7.8.8.0. Tomographic tilt-series were recorded using SerialEM 3.8.5. The simulation code is available at https://github.com/luiree/TipPhase. |
|---|---|
| Data analysis | Fluorescence microscopy images were analyzed using Fiji 2.0<br>Tomograms were resonstructed using IMOD v. 4.9.2. Segmentation was performed using EMAN v 2,2.<br>Python scripts used for splitting of movie frames, reconstruction of even and odd tomographic volumes, training data generation, model training and denoising are available at https://github.com/NemoAndrea/cryoCARE-hpc04. |

For manuscripts utilizing custom algorithms or software that are central to the research but not yet described in published literature, software must be made available to editors and reviewers. We strongly encourage code deposition in a community repository (e.g. GitHub). See the Nature Portfolio guidelines for submitting code & software for further information.

## Data

Policy information about availability of data

All manuscripts must include a data availability statement. This statement should provide the following information, where applicable:
- Accession codes, unique identifiers, or web links for publicly available datasets
- A description of any restrictions on data availability
- For clinical datasets or third party data, please ensure that the statement adheres to our policy

Tomography data presented in Figure 2 are available from EMDB using the following accession codes: microtubule plus-end in presence of Tip1, Tea2 and Mal3 (EMD-14110), microtubule minus-end in presence of Tip1, Tea2 and Mal3 (EMD14111), microtubule plus-end in presence of Mal3 (EMD-1408), microtubule minus-end in presence of Mal3 (EMD-14109), microtubule plus-end in absence of additional proteins (EMD-14106), microtubule minus-end in absence of additional

# Field-specific reporting

Please select the one below that is the best fit for your research. If you are not sure, read the appropriate sections before making your selection.

☒ Life sciences ☐ Behavioural & social sciences ☐ Ecological, evolutionary & environmental sciences

For a reference copy of the document with all sections, see nature.com/documents/nr-reporting-summary-flat.pdf

# Life sciences study design

All studies must disclose on these points even when the disclosure is negative.

| | |
|---|---|
| Sample size | No sample size pre-determination was used. Reported data were generated from at least three independent experimental replicates. |
| Data exclusions | No data were excluded from analysis |
| Replication | Reported data were generated from at least three independent experimental replicates. All data were included in the analysis reported in the paper |
| Randomization | No randomization was performed as it doesn't apply to our study |
| Blinding | No blinding was performed as it doesn't apply to our study |

# Reporting for specific materials, systems and methods

We require information from authors about some types of materials, experimental systems and methods used in many studies. Here, indicate whether each material, system or method listed is relevant to your study. If you are not sure if a list item applies to your research, read the appropriate section before selecting a response.

## Materials & experimental systems

| n/a | Involved in the study |
|---|---|
| ☒ ☐ | Antibodies |
| ☒ ☐ | Eukaryotic cell lines |
| ☒ ☐ | Palaeontology and archaeology |
| ☒ ☐ | Animals and other organisms |
| ☒ ☐ | Human research participants |
| ☒ ☐ | Clinical data |
| ☒ ☐ | Dual use research of concern |

## Methods

| n/a | Involved in the study |
|---|---|
| ☒ ☐ | ChIP-seq |
| ☒ ☐ | Flow cytometry |
| ☒ ☐ | MRI-based neuroimaging |

