## [Peer Review File · Nature Cell Biology]

Peer Review Information

Journal: Nature Cell Biology

Manuscript Title: Multivalent interactions facilitate motor-dependent protein accumulation at growing microtubule plus ends

Corresponding author name(s): Professor Marileen Dogterom

Editorial Notes:

Reviewer Comments & Decisions:

Decision Letter, initial version:

*Please delete the link to your author homepage if you wish to forward this email to co-authors.

Dear Professor Dogterom,

I hope you are well and I apologize for the very long delay in the peer review of your submission.

Your manuscript, "Multivalent interactions facilitate motor-dependent protein accumulation at growing microtubule plus ends", has now been seen by 3 referees, who are experts in microtubules (referee 1); biomolecular condensation (referee 2); and CryoET (referee 3). As you will see from their comments (attached below) they find this work of potential interest, but have raised substantial concerns, which in our view would need to be addressed with considerable revisions before we can consider publication in Nature Cell Biology.

Nature Cell Biology editors discuss the referee reports in detail within the editorial team, including the chief editor, to identify key referee points that should be addressed with priority, and requests that are overruled as being beyond the scope of the current study. To guide the scope of the revisions, I have listed these points below. We are committed to providing a fair and constructive peer-review process, so please feel free to contact me if you would like to discuss any of the referee comments further.

In particular, it would be essential to:

A) Test hypotheses generated from the modelling predictions (Reviewer #2)

B) Assess potential differences in biomolecular condensation in vitro as opposed to those on microtubules (Reviewer #2)

C) Characterize biomolecular condensation behaviour on microtubule behaviour with potential to be motor-activity-driven (Reviewer #2)

D) Provide non-denoised CryoET images (Reviewer #3)

E) All other referee concerns pertaining to strengthening existing data, providing controls, methodological details, clarifications and textual changes, should also be addressed.

F) Finally please pay close attention to our guidelines on statistical and methodological reporting (listed below) as failure to do so may delay the reconsideration of the revised manuscript. In particular please provide:

We would be happy to consider a revised manuscript that would satisfactorily address these points, unless a similar paper is published elsewhere, or is accepted for publication in Nature Cell Biology in the meantime.

- ensure that it conforms to our format instructions and publication policies (see below and

<https://www.nature.com/nature/for-authors>).

- provide a point-by-point rebuttal to the full referee reports verbatim, as provided at the end of this letter.

- provide the completed Reporting Summary (found here <https://www.nature.com/documents/nr-reporting-summary.pdf>). This is essential for reconsideration of the manuscript will be available to editors and referees in the event of peer review. For more information see <http://www.nature.com/authors/policies/availability.html> or contact me.

When submitting the revised version of your manuscript, please pay close attention to our [href="https://www.nature.com/nature-research/editorial-policies/image-integrity">Digital Image Integrity Guidelines](https://www.nature.com/nature-research/editorial-policies/image-integrity). and to the following points below:

Nature Cell Biology is committed to improving transparency in authorship. As part of our efforts in this direction, we are now requesting that all authors identified as 'corresponding author' on published papers create and link their Open Researcher and Contributor Identifier (ORCID) with their account on the Manuscript Tracking System (MTS), prior to acceptance. ORCID helps the scientific community achieve unambiguous attribution of all scholarly contributions. You can create and link your ORCID from the home page of the MTS by clicking on 'Modify my Springer Nature account'. For more information please visit www.springernature.com/orcid.

This journal strongly supports public availability of data. Please place the data used in your paper into a public data repository, or alternatively, present the data as Supplementary Information. If data can only be shared on request, please explain why in your Data Availability Statement, and also in the correspondence with your editor. Please note that for some data types, deposition in a public repository is mandatory - more information on our data deposition policies and available repositories appears below.

[Redacted]

We would like to receive a revised submission within six months.

We hope that you will find our referees' comments, and editorial guidance helpful. Please do not hesitate to contact me if there is anything you would like to discuss.

Best wishes,

Daryl Jason David

Daryl J.V. David, PhD

Senior Editor, Nature Cell Biology
Consulting Editor, Nature Communications
Nature Portfolio

Heidelberger Platz 3, 14197 Berlin, Germany
Email: daryl.david@nature.com
ORCID: <https://orcid.org/0000-0002-9253-4805>

Reviewers' Comments:

Reviewer #1:

Remarks to the Author:

In their manuscript, Maan et al. determine the role of liquid-liquid phase separation in the formation of a structure known as the microtubule +TIP comet. This comet is assembled by the "master regulator" EB1 (and its homologs in different species) and assembles a number of different +TIP proteins at the growing microtubule ends. How the different proteins interact to form the +TIP comets has so far not been fully understood.

The authors use components of the fission yeast +TIP tracking system to characterise their assembly of assemblies and test the hypothesis of multivalent interactions leading of liquid-liquid phase separation in this process. The work is a powerful combination of in vitro reconstitution experiments with purified proteins, cryo-EM approaches and theoretical modelling.

The authors first demonstrate that Mal3, the EB-homolog, Tip1, CLIP170 homolog, and Tea2, the kinesin-7 homolog together form condensates either with or without microtubules. The condensates are liquid, can coat microtubules, and are transported by Tea2 motors towards the plus-ends of microtubules, where they form the characteristic comets. These observations were all made under crowding conditions in cell-free in-vitro reconstitutions from recombinant proteins purified from *E. coli*. To determine the role of the crowding agent that was necessary to see comets in light-microscopy based reconstitution, the authors turned to cryo-EM. After solving the cumbersome problem of non-specific protein attachment to the EM grids, the authors show that the comets formed under non-crowded conditions are similar to the ones observed with crowding agents, but are more loosely structured. This explains the need to crowding agents in the fluorescent-based assays, however proves that the structures do also assemble in the absence of crowding agents.

The authors next report the striking observation that the 3 components do not accumulate stoichiometrically at the microtubule plus end changing the concentration of the motor Tea2 did not affect the concentration of Mal3, but did disproportionately affect the Tip1 concentration. This non-stoichiometric accumulation of the 3 proteins is another indication that they do not form defined protein complexes, but rather assemble by multivalent interactions.

In the next step the authors determine the role of different domains of Mal3 in the observed comet-forming behaviour. Strikingly, all deletions that prevented +TIP tracking of Mal3 also prevented

droplet formation. The importance of different protein domains for protein-protein interactions was further dissected with guest-host assays. The sum of all assays performed leads the authors to the conclusion that the sum of different molecular interactions coordinated by Mal3 are essential for the formation of the +TIP comet.

Finally, the authors use molecular modelling to theoretically describe the +TIP tracking of the 3 proteins.

Overall, the manuscript is well-written, the experiments are described in detail to allow the reader a good understanding of the experimental settings and the results. Figures are also of high quality. However, reading the paper as a whole it becomes obvious that different chapters have been written in a different style, which is most obvious for the modelling chapter that is by far the longest and most detailed part of the result section. This inconsistent structure of the manuscript makes it difficult to follow at places, thus some streamlining of the text would be largely profitable for the storyline. Similarly, the introduction of the paper could be more concise. There is an extensive description of the results that takes almost an entire page, which is not necessary given that similar information is given in the results and the discussion. Thus, one of the central points that needs to be addressed is the writing: the manuscript could profit from being more concise, linear and written in a form easy accessible for a general public.

Reviewer #2:

Remarks to the Author:

In the article entitled "Multivalent interactions facilitate motor-dependent protein accumulation at growing microtubule ends," Maan, Reese, Volkov and colleagues investigate the interactions between Mal3, Tea2, and Tip1 that give rise to condensates that form and localize at the plus end of microtubules. Using a combination of in vitro reconstitution, cryoET, and computational modeling, the authors identified two intrinsically disordered regions in Mal3; both regions were required for Tip1 localization at the plus end, but only one region was necessary for condensate (comet) formation on microtubules. The authors conclude that the combination of motor activity and multivalent interactions between Mal3, Tea2, and Tip1 drive their condensation at the plus ends of microtubules.

This article provides stunning structural insight into the Mal3 and Mal3, Tip1, Tea2 network of proteins localized to the plus-end of microtubules. Their work to understand the phase separation of Mal3 in the presence of crowding agents provides important insight into its phase behavior. In addition to these observations, the liquid like behavior of plus-end condensates is extremely important for the fields of phase separation and cytoskeletal biology. Finally, the theoretical model provides insight into a potential mechanism that can explain the experimental data presented in this manuscript. Overall, this is an excellent manuscript that provides a unique picture into mechanisms by which plus-end condensates can be regulated. However, as will be discussed below, it would be beneficial for the authors to test their model using an experimental system in which they perturb both model and experiments to confirm that their model accurately describes the biological system. Otherwise, it could be argued that the presented model is simply a data-fitting algorithm rather than a bona fide model that can be used to predict plus-end condensate dynamics with other binding partners. If this concern and other minor comments can be addressed, this manuscript is a strong candidate for publication in Nature Cell Biology.

Major Comment:

1) As noted above, the authors have set up their model as a mathematical description rather than a tool to predict testable experimental manipulations. It would be of great benefit to test some of their model predictions, such as motor slowdown. The authors should perform experiments with Tea2 that has either high or low velocity along the microtubule to test whether the prediction that motor speed will alter Mal3 condensation at the plus-end.

Minor Comments:

- 1) In Fig 1H and video S2, it appears that condensates wet the microtubules only in the plus-end direction. This seems to suggest that condensate wetting is at least partially motor-driven rather than passive wetting as seen by condensates made by non-active matter. The authors should note this more explicitly in lines 153-156.
- 2) In lines 167-170, the authors posit that Plateau-Rayleigh instability is responsible for the observations in Figure 1I. This would require that the condensate on the microtubule is a single condensate that only forms discrete droplets after falling off the end of the microtubule. Another potential explanation for these observations is that discrete droplets exist on the microtubule, fall off the end of the moving microtubule, and remain discrete droplets on the substrate. Is there any additional evidence that can be used to support Plateau-Rayleigh instability?
- 3) In lines 226-230, the authors describe the macroscale difference between in vitro condensates and condensates on the microtubule. But what about the structure on the microscale? Based on the CryoET images, it appears that the microtubule-bound networks are like the larger scale in vitro networks within the condensates. Could this be possible? If this is the case, what are the biological implications or condensates with the same internal structure, yet scaled at different sizes? Would the authors expect the experiments with the larger condensates to be predictive of the smaller condensates on the microtubules if the internal environment of condensates is similar?
- 4) Starting in line 303, the authors should use the common "scaffold / client" terminology (See PMID 28225081 for further details).

Reviewer #3:

Remarks to the Author:

This is a very nicely designed study to gain clarity on the increased concentrations and phase separation of several microtubule plus-end proteins. The study uses a variety of methods including fluorescence, CryoET and simulations. Overall the methodology seems well designed and properly carried out, and the conclusions justified by the data. This study is likely to be of interest primarily to microtubule specialists, but as a non-specialist, I still found it interesting to read. I believe the publication to be suitable for publication, with two critical modifications. I do not believe either of these changes would require re-review of the manuscript.

First, I cannot find figure captions for the supplementary figures anywhere. While I could figure out what was going on well enough for the review, clearly this needs to be addressed.

Second, the CryoCARE method used for denoising the tomograms is quite new, and not yet widely accepted in the community. Deep learning based denoising methods can modify raw data in unpredictable and potentially undesirable ways. Certainly the procedure is acceptable for purposes of making annotation easier, if the annotations are then presented on top of the original data. However, unlike standard operations like low-pass filters which are well understood, and cannot remove discrete objects, presenting denoised data as the primary result without also presenting the original data without denoising is not acceptable. I do believe it is very unlikely that the presentation of reconstructions without denoising would alter any of the conclusions in this particular study, but side-by-side presentation of all presented data with/without denoising is absolutely necessary.

REFERENCES – are limited to a total of 70 for Articles, Resources, Technical Reports; and 40 for Letters. This includes references in the main text and Methods combined. References must be numbered sequentially as they appear in the main text, tables and figure legends and Methods and must follow the precise style of Nature Cell Biology references. References only cited in the Methods

should be numbered consecutively following the last reference cited in the main text. References only associated with Supplementary Information (e.g. in supplementary legends) do not count toward the total reference limit and do not need to be cited in numerical continuity with references in the main text. Only published papers can be cited, and each publication cited should be included in the numbered reference list, which should include the manuscript titles. Footnotes are not permitted.

Methods should be written concisely, but should contain all elements necessary to allow interpretation and replication of the results. As a guideline, Methods sections typically do not exceed 3,000 words. The Methods should be divided into subsections listing reagents and techniques. When citing previous methods, accurate references should be provided and any alterations should be noted. Information must be provided about: antibody dilutions, company names, catalogue numbers and clone numbers for monoclonal antibodies; sequences of RNAi and cDNA probes/primers or company names and catalogue numbers if reagents are commercial; cell line names, sources and information on cell line identity and authentication. Animal studies and experiments involving human subjects must be reported in detail, identifying the committees approving the protocols. For studies involving human subjects/samples, a statement must be included confirming that informed consent was obtained. Statistical analyses and information on the reproducibility of experimental results should be provided in a section titled "Statistics and Reproducibility".

All Nature Cell Biology manuscripts submitted on or after March 21 2016 must include a Data availability statement as a separate section after Methods but before references, under the heading "Data Availability". For Springer Nature policies on data availability see <http://www.nature.com/authors/policies/availability.html>; for more information on this particular policy see <http://www.nature.com/authors/policies/data/data-availability-statements-data-citations.pdf>. The Data availability statement should include:

- Accession codes for primary datasets (generated during the study under consideration and designated as "primary accessions") and secondary datasets (published datasets reanalysed during the study under consideration, designated as "referenced accessions"). For primary accessions data should be made public to coincide with publication of the manuscript. A list of data types for which submission to community-endorsed public repositories is mandated (including sequence, structure, microarray, deep sequencing data) can be found here <http://www.nature.com/authors/policies/availability.html#data>.
- Unique identifiers (accession codes, DOIs or other unique persistent identifier) and hyperlinks for datasets deposited in an approved repository, but for which data deposition is not mandated (see here for details <http://www.nature.com/sdata/data-policies/repositories>).
- At a minimum, please include a statement confirming that all relevant data are available from the authors, and/or are included with the manuscript (e.g. as source data or supplementary information), listing which data are included (e.g. by figure panels and data types) and mentioning any restrictions on availability.
- If a dataset has a Digital Object Identifier (DOI) as its unique identifier, we strongly encourage including this in the Reference list and citing the dataset in the Methods.

We recommend that you upload the step-by-step protocols used in this manuscript to the Protocol Exchange. More details can found at www.nature.com/protocolexchange/about.

All imaging data should be accompanied by scale bars, which should be defined in the legend. Cropped images of gels/blots are acceptable, but need to be accompanied by size markers, and to retain visible background signal within the linear range (i.e. should not be saturated). The boundaries of panels with low background have to be demarked with black lines. Splicing of panels should only be considered if unavoidable, and must be clearly marked on the figure, and noted in the legend with a statement on whether the samples were obtained and processed simultaneously. Quantitative comparisons between samples on different gels/blots are discouraged; if this is unavoidable, it should only be performed for samples derived from the same experiment with gels/blots were processed in parallel, which needs to be stated in the legend.

The total number of Supplementary Figures (not including the “unprocessed scans” Supplementary Figure) should not exceed the number of main display items (figures and/or tables (see our Guide to Authors and March 2012 editorial <http://www.nature.com/ncb/authors/submit/index.html#suppinfo>; <http://www.nature.com/ncb/journal/v14/n3/index.html#ed>). No restrictions apply to Supplementary Tables or Videos, but we advise authors to be selective in including supplemental data.

Each Supplementary Figure should be provided as a single page and as an individual file in one of our accepted figure formats and should be presented according to our figure guidelines (see above). Supplementary Tables should be provided as individual Excel files. Supplementary Videos should be provided as .avi or .mov files up to 50 MB in size. Supplementary Figures, Tables and Videos must be

accompanied by a separate Word document including titles and legends.

GUIDELINES FOR EXPERIMENTAL AND STATISTICAL REPORTING

REPORTING REQUIREMENTS – We are trying to improve the quality of methods and statistics reporting in our papers. To that end, we are now asking authors to complete a reporting summary that collects information on experimental design and reagents. The Reporting Summary can be found here <https://www.nature.com/documents/nr-reporting-summary.pdf> If you would like to reference the guidance text as you complete the template, please access these flattened versions at <http://www.nature.com/authors/policies/availability.html>.

Author Rebuttal to Initial comments

Reviewers' Comments:

We would like to thank all three reviewers for their constructive comments, which helped us improve our manuscript. In addition to rewriting parts of the text (see for details below), we added new data to

Figure 2 and Supplementary Fig. S3 (previously Fig. S2) (both a new condition and increased statistics for previously shown data). We also added a new Supplementary Fig. S2 in response to reviewer 3. Below we respond in detail to each of the reviewers' comments.

Reviewer #1:

Remarks to the Author:

In their manuscript, Maan et al. determine the role of liquid-liquid phase separation in the formation of a structure known as the microtubule +TIP comet. This comet is assembled by the “master regulator” EB1 (and its homologs in different species) and assembles a number of different +TIP proteins at the growing microtubule ends. How the different proteins interact to form the +TIP comets has so far not been fully understood.

The authors use components of the fission yeast +TIP tracking system to characterise their assembly of assemblies and test the hypothesis of multivalent interactions leading of liquid-liquid phase separation in this process. The work is a powerful combination of in vitro reconstitution experiments with purified proteins, cryo-EM approaches and theoretical modelling.

The authors first demonstrate that Mal3, the EB-homolog, Tip1, CLIP170 homolog, and Tea2, the kinesin-7 homolog together form condensates either with or without microtubules. The condensates are liquid, can coat microtubules, and are transported by Tea2 motors towards the plus-ends of microtubules, where they form the characteristic comets. These observations were all made under crowding conditions in cell-free in-vitro reconstitutions from recombinant proteins purified from E. coli. To determine the role of the crowding agent that was necessary to see comets in light-microscopy based reconstitution, the authors turned to cryo-EM. After solving the cumbersome problem of non-specific protein attachment to the EM grids, the authors show that the comets formed under non-crowded conditions are similar to the ones observed with crowding agents, but are more loosely structured. This explains the need to crowding agents in the fluorescent-based assays, however proves that the structures do also assemble in the absence of crowding agents.

The authors next report the striking observation that the 3 components do not accumulate stoichiometrically at the microtubule plus end changing the concentration of the motor Tea2 did not affect the concentration of Mal3, but did disproportionately affect the Tip1 concentration. This non-stoichiometric accumulation of the 3 proteins is another indication that they do not form defined protein complexes, but rather assemble by multivalent interactions.

In the next step the authors determine the role of different domains of Mal3 in the observed comet-forming behaviour. Strikingly, all deletions that prevented +TIP tracking of Mal3 also prevented droplet formation. The importance of different protein domains for protein-protein interactions was further

dissected with guest-host assays. The sum of all assays performed leads the authors to the conclusion that the sum of different molecular interactions coordinated by Mal3 are essential for the formation of the +TIP comet.

Finally, the authors use molecular modelling to theoretically describe the +TIP tracking of the 3 proteins.

Overall, the manuscript is well-written, the experiments are described in detail to allow the reader a good understanding of the experimental settings and the results. Figures are also of high quality.

However, reading the paper as a whole it becomes obvious that different chapters have been written in a different style, which is most obvious for the modelling chapter that is by far the longest and most detailed part of the result section. This inconsistent structure of the manuscript makes it difficult to follow at places, thus some streamlining of the text would be largely profitable for the storyline.

Similarly, the introduction of the paper could be more concise. There is an extensive description of the results that takes almost an entire page, which is not necessary given that similar information is given in the results and the discussion. Thus, one of the central points that needs to be addressed is the writing: the manuscript could profit from being more concise, linear and written in a form easy accessible for a general public.

We thank the reviewer for the positive comments. We appreciate that the writing can be improved and have thus significantly shortened both the introduction and the modeling section. See also the response to the comment about modeling by reviewer 2 below.

Reviewer #2:

Remarks to the Author:

In the article entitled “Multivalent interactions facilitate motor-dependent protein accumulation at growing microtubule ends,” Maan, Reese, Volkov and colleagues investigate the interactions between Mal3, Tea2, and Tip1 that give rise to condensates that form and localize at the plus end of microtubules. Using a combination of in vitro reconstitution, cryoET, and computational modeling, the authors identified two intrinsically disordered regions in Mal3; both regions were required for Tip1 localization at the plus end, but only one region was necessary for condensate (comet) formation on microtubules. The authors conclude that the combination of motor activity and multivalent interactions between Mal3, Tea2, and Tip1 drive their condensation at the plus ends of microtubules.

This article provides stunning structural insight into the Mal3 and Mal3, Tip1, Tea2 network of proteins

localized to the plus-end of microtubules. Their work to understand the phase separation of Mal3 in the presence of crowding agents provides important insight into its phase behavior. In addition to these observations, the liquid like behavior of plus-end condensates is extremely important for the fields of phase separation and cytoskeletal biology. Finally, the theoretical model provides insight into a potential mechanism that can explain the experimental data presented in this manuscript. Overall, this is an excellent manuscript that provides a unique picture into mechanisms by which plus-end condensates can be regulated. However, as will be discussed below, it would be beneficial for the authors to test their model using an experimental system in which they perturb both model and experiments to confirm that their model accurately describes the biological system. Otherwise, it could be argued that the presented model is simply a data-fitting algorithm rather than a bona fide model that can be used to predict plus-end condensate dynamics with other binding partners. If this concern and other minor comments can be addressed, this manuscript is a strong candidate for publication in Nature Cell Biology.

Major Comment:

1) As noted above, the authors have set up their model as a mathematical description rather than a tool to predict testable experimental manipulations. It would be of great benefit to test some of their model predictions, such as motor slowdown. The authors should perform experiments with Tea2 that has either high or low velocity along the microtubule to test whether the prediction that motor speed will alter Mal3 condensation at the plus-end.

We appreciate the comment that testable model predictions and a direct quantitative comparison between model and experiments would in principle be desirable. We however run into the problem that our experimental system is highly complex: plus-end accumulation of the three MPET network components (Mal3, Tea2 and Tip1) is a result of both motor-driven transport towards the plus end and autonomous interaction of Mal3 with the comet region near the growing MT ends. Furthermore, varying the concentration of each of the components in the assay is likely to change the balance of complex formation both in solution and on the microtubule lattice making straightforward predictions about the resulting effects on both lattice coverage and end-accumulation non-trivial.

Since we do not have enough information to model the complete system and make direct quantitative comparisons, we feel we can realistically only use modelling to gain intuition on how different assumptions about motor (Tea2) and cargo (Tip1) behavior may affect end-accumulation, complementing a series of previously published models of (single component) motor traffic jams (see e.g. Leduc et al, 2012). We specifically focus on the accumulation of the cargo Tip1 which experimentally appears to increase with motor concentration in a non-stoichiometric way (Fig. 3E). For Mal3 it is known that adding motors is not necessary to obtain end-accumulation and we furthermore show that in the

presence of motors the amount of Mal3 at the tip does not increase when more motors are added (Fig. 3D). Mal3 is therefore not explicitly included in the model. Instead, its effect is indirectly included when we assume cargo molecules to form multi-component clusters.

Using our modeling efforts, we first of all argue that slow-down of motors near microtubule ends (due to effects that are specific for the comet region such as a different nucleotide state and/or molecular crowding) is a sufficient condition to accumulate cargo at microtubule ends, even if motors run off freely when they reach the very tip of the microtubule. This is a relevant feature of our experiments and complements previous models where it was shown that end-accumulation (or “spikes”) may also (or in addition) result from a reduced motor detachment rate at the microtubule end (and we don’t exclude that both effects play a role in our system). Changing the motor velocity on the whole microtubule as suggested by the reviewer has no effect on this phenomenon, since it is the change in motor velocity when reaching the comet region near the growing end that is responsible for accumulation. Removing the end-specific change in velocity can be achieved by growing microtubules with a non-hydrolysable analogue such as GTPgammaS (Fig. S6A). Indeed, no clear end-accumulation is observed under these conditions. Note however, that the motor density is very high on these filaments due to increased lattice affinity which may lead to the absence of “spikes” that would result from a potential end-specific motor detachment rate (Leduc *al*, Fig. 4; see also our additional experiments below).

Nevertheless, to explore in more detail the effect of a boundary between two microtubule regions on motor/cargo behavior, we performed additional experiments where we created microtubule lattices composed of three different regions with three different nucleotide states: we used stabilized GMPCPP seeds to first nucleate dynamic microtubules in the presence of GTP (as elsewhere in our paper). We then replaced the solution with a solution containing tubulin and GTPgammaS to grow long stable microtubule regions at both the plus- and minus-ends of the microtubules. We then added our three MPET components as before and followed the behavior of Tip1 when reaching the boundaries between these various segments leading to the following observations (see Figure below):

- The landing rate of motor/cargo complexes was highest on the gammaS regions (consistent with our observations in Fig. 6SA), although the overall density on gammaS was lower in these new experiments (potentially because of a different distribution of the different components between different possible complexes in solution and on the different microtubule lattices).
- While not forming steady comets with roughly constant intensity as observed in our normal assays (Fig. 1B), Tip1 clusters were able to (transiently) accumulate at growing ends even though there was no boundary between lattice types near the microtubule ends in these experiments. We assume this to be the result of an end-specific motor off-rate (see Leduc, 2012).
- Tip1 was also sometimes observed to accumulate at the minus ends of the GMPCPP seeds, but only when a GDP segment at the minus end was missing (bottom row in figure below). This

shows that the boundary between a gammaS region (which mimics the GDP-Pi comet region on a normal growing MT) to a GMPCPP region (which mimics the very GTP tip of growing MT) may also contribute to accumulation at growing microtubule ends. The same was not observed when a GDP segment was present between the gammaS segment and the seed (top row in figure below).

Figure: two-color kymographs showing transport of labeled Tip1 (turquoise) on segmented microtubules. Only the GDP segments of the microtubules are labeled (red). GMPCPP seeds (dark areas) may be flanked by GDP segments on their plus and/or minus ends.

These observations illustrate the different effects that boundaries between different microtubule lattice regions as well as lattice ends may have on motor/cargo behavior, further emphasizing the complexity of the experimental system.

In addition, we were interested in finding a simple mechanism that could explain the non-linear motor-dependence of Tip1 accumulation at the microtubule end. Using our model, we show that simply allowing cargo to form clusters does not lead to enhanced accumulation (compare Fig. 6B to Fig. 6C). Instead, a clustering-dependent decrease in the motor off-rate (or a clustering-dependent increase in the residence time) does lead to non-linear effects (Fig. 6DEF). Note that this is only a first natural step in increasing the complexity of the model which gives us insight in a possible mechanism behind our experimental observations. Additional complexity could be incorporated by also making the cargo off-rate clustering-dependent which will add additional non-linear effects. However, as explained above, we restrict ourselves to moderate changes to existing models as the full complexity of the system is beyond reach of our modeling efforts.

Realizing all these limitations of our modelling efforts, one may wonder why to include them at all. We firmly believe however, that simple extensions of existing models that include relevant features of real experimental systems contributes to the field in a broader sense, as they may also help understand experimental systems other than the specific one described here.

To better explain the merit of our modelling efforts and what it can and cannot predict given the complexity of our experimental system, we have rewritten (and shortened) this section of the paper (also in response to the comment made by reviewer 1).

Minor Comments:

1) In Fig 1H and video S2, it appears that condensates wet the microtubules only in the plus-end direction. This seems to suggest that condensate wetting is at least partially motor-driven rather than passive wetting as seen by condensates made by non-active matter. The authors should note this more explicitly in lines 153-156.

We have made the suggested change in the text

2) In lines 167-170, the authors posit that Plateau-Rayleigh instability is responsible for the observations

in Figure 11. This would require that the condensate on the microtubule is a single condensate that only forms discrete droplets after falling off the end of the microtubule. Another potential explanation for these observations is that discrete droplets exist on the microtubule, fall off the end of the moving microtubule, and remain discrete droplets on the substrate. Is there any additional evidence that can be used to support Plateau-Rayleigh instability?

In our videos (such as video S2) we observe that the motor-driven condensates increase in size over time while maintaining an apparent spherical shape. We interpret this as evidence that a single condensate is formed.

3) In lines 226-230, the authors describe the macroscale difference between in vitro condensates and condensates on the microtubule. But what about the structure on the microscale? Based on the CryoET images, it appears that the microtubule-bound networks are like the larger scale in vitro networks within the condensates. Could this be possible? If this is the case, what are the biological implications or condensates with the same internal structure, yet scaled at different sizes? Would the authors expect the experiments with the larger condensates to be predictive of the smaller condensates on the microtubules if the internal environment of condensates is similar?

We thank the reviewer for this observation and suggestions. The internal (microscale) structure of both condensates and comets in Fig. 2) becomes clearly visible only after the denoising procedure that removes any high-resolution information from the tomograms (see also new Supplementary Fig. S2). We therefore hesitate to make conclusions about the microscale structure of the microtubule-bound comets in comparison with condensates formed in presence of crowding agents. However, to illustrate the difference between these two structures within one sample, we performed additional experiments where we first polymerized microtubules with end-tracking Mal3/Tip1/Tea2 in the absence of PEG, and then added PEG-35k + Mal3/Tip1/Tea2 (no additional tubulin) to the same sample. We also generated more data in absence of PEG to increase the statistics. These new data are added in Figure 2 and Supplementary Figures S2 (new) and S3 (former S2). As we write in the updated section of the Results, on a subset of microtubule ends we observe extended droplet-like structures, often in addition to pre-formed comets that resemble Mal3/Tip1/Tea2 comets formed in absence of PEG. On other microtubule ends, we only observe the (smaller size) comet structures. These data show that the comets remain network-like structures that are not as homogeneous in density as the larger droplets, even in the presence of PEG. Potentially, these structures need more material to become droplet-like. The small-scale (10-30nm) internal structure does however look similar.

4) Starting in line 303, the authors should use the common “scaffold / client” terminology (See PMID

28225081 for further details).

We thank the reviewer and made the suggested changes.

Reviewer #3:

Remarks to the Author:

This is a very nicely designed study to gain clarity on the increased concentrations and phase separation of several microtubule plus-end proteins. The study uses a variety of methods including fluorescence, CryoET and simulations. Overall the methodology seems well designed and properly carried out, and the conclusions justified by the data. This study is likely to be of interest primarily to microtubule specialists, but as a non-specialist, I still found it interesting to read. I believe the publication to be suitable for publication, with two critical modifications. I do not believe either of these changes would require re-review of the manuscript.

First, I cannot find figure captions for the supplementary figures anywhere. While I could figure out what was going on well enough for the review, clearly this needs to be addressed.

We apologize for this omission which occurred due to a mistake during the submission process. The supplementary figure captions are now properly included in the Supplementary Information file.

Second, the CryoCARE method used for denoising the tomograms is quite new, and not yet widely accepted in the community. Deep learning based denoising methods can modify raw data in unpredictable and potentially undesirable ways. Certainly the procedure is acceptable for purposes of making annotation easier, if the annotations are then presented on top of the original data. However, unlike standard operations like low-pass filters which are well understood, and cannot remove discrete objects, presenting denoised data as the primary result without also presenting the original data without denoising is not acceptable. I do believe it is very unlikely that the presentation of reconstructions without denoising would alter any of the conclusions in this particular study, but side-by-side presentation of all presented data with/without denoising is absolutely necessary.

We thank the reviewer for the suggestion to emphasize the validation of results obtained using cryoCARE. In the new Supplementary Figure S2 we provide slices from denoised and non-denoised tomograms side by side with the same slices taken from tomograms processed using nonlinear anisotropic diffusion algorithm (NAD). Raw, non-denoised tomograms are also uploaded to EMDB and will be released upon the publication of this paper.

Decision Letter, first revision:

Our ref: NCB-A46790A

12th July 2022

Dear Dr. Dogterom,

I hope you are well and I apologize for the delay. Thank you for submitting your revised manuscript "Multivalent interactions facilitate motor-dependent protein accumulation at growing microtubule plus ends" (NCB-A46790A). It has now been seen by the original referees and their comments are below. The reviewers find that the paper has improved in revision, and therefore we'll be happy in principle to publish it in Nature Cell Biology, pending minor revisions to satisfy the referees' final requests and to comply with our editorial and formatting guidelines.

Thank you again for your interest in Nature Cell Biology Please do not hesitate to contact me if you have any questions.

Sincerely,
Daryl

Daryl J.V. David, PhD

Senior Editor, Nature Cell Biology
Consulting Editor, Nature Communications
Nature Portfolio

Heidelberger Platz 3, 14197 Berlin, Germany
Email: daryl.david@nature.com
ORCID: <https://orcid.org/0000-0002-9253-4805>

Reviewer #1 (Remarks to the Author):

In the revised version of the manuscript, the authors have addressed my concerns and suggestions. The paper could therefore be published in the current form.

Reviewer #2 (Remarks to the Author):

The authors have adequately addressed the concerns of this reviewer. This reviewer appreciates the time and energy the authors spent in 1) responding to this reviewer's critiques and 2) performing additional experiments to address this reviewer's concerns. This reviewer agrees with the authors that the inclusion of their model is important and warranted, as it is a first step towards the creation of a more comprehensive model and can be used to predict and understand any number of experimental outcomes in the future. The authors' brief discussion of their model is helpful for their goals and the results of the model. With the improvements made to the manuscript, this reviewer supports its publication in Nature Cell Biology.

Minor Comments:

- 1) If space allows a concluding / summary sentence in the abstract would be helpful.
- 2) In line 61, 'unstructured' should be 'disordered', as all proteins have some structure, whether primary, secondary, etc.
- 3) In line 282, 'truncates' should be 'truncations'.

Reviewer #3 (Remarks to the Author):

The revised manuscript addresses the issues I raised in the original review, and I believe it is suitable for publication. I have no further comments to add.

Decision Letter, final checks:

Our ref: NCB-A46790A

4th August 2022

Dear Dr. Dogterom,

Thank you for your patience as we've prepared the guidelines for final submission of your Nature Cell Biology manuscript, "Multivalent interactions facilitate motor-dependent protein accumulation at growing microtubule plus ends" (NCB-A46790A). Please carefully follow the step-by-step instructions provided in the attached file, and add a response in each row of the table to indicate the changes that you have made. Please also check and comment on any additional marked-up edits we have proposed within the text. Ensuring that each point is addressed will help to ensure that your revised manuscript can be swiftly handed over to our production team.

We would like to start working on your revised paper, with all of the requested files and forms, as soon as possible (preferably within one week). Please get in contact with us if you anticipate delays.

If you have not done so already, please alert us to any related manuscripts from your group that are

under consideration or in press at other journals, or are being written up for submission to other journals (see: <https://www.nature.com/nature-research/editorial-policies/plagiarism#policy-on-duplicate-publication> for details).

In recognition of the time and expertise our reviewers provide to Nature Cell Biology's editorial process, we would like to formally acknowledge their contribution to the external peer review of your manuscript entitled "Multivalent interactions facilitate motor-dependent protein accumulation at growing microtubule plus ends". For those reviewers who give their assent, we will be publishing their names alongside the published article.

Nature Cell Biology offers a Transparent Peer Review option for new original research manuscripts submitted after December 1st, 2019. As part of this initiative, we encourage our authors to support increased transparency into the peer review process by agreeing to have the reviewer comments, author rebuttal letters, and editorial decision letters published as a Supplementary item. When you submit your final files please clearly state in your cover letter whether or not you would like to participate in this initiative. Please note that failure to state your preference will result in delays in accepting your manuscript for publication.

Cover suggestions

As you prepare your final files we encourage you to consider whether you have any images or illustrations that may be appropriate for use on the cover of Nature Cell Biology.

Nature Cell Biology has now transitioned to a unified Rights Collection system which will allow our Author Services team to quickly and easily collect the rights and permissions required to publish your work. Approximately 10 days after your paper is formally accepted, you will receive an email in providing you with a link to complete the grant of rights. If your paper is eligible for Open Access, our Author Services team will also be in touch regarding any additional information that may be required to arrange payment for your article.

Please note that *Nature Cell Biology* is a Transformative Journal (TJ). Authors may publish their research with us through the traditional subscription access route or make their paper immediately open access through payment of an article-processing charge (APC). Authors will not be required to make a final decision about access to their article until it has been accepted. Find out more about

Transformative Journals

[Redacted]

Best regards,

Nyx Hills
Staff
Nature Cell Biology

On behalf of

Daryl J.V. David, PhD

Senior Editor, Nature Cell Biology
Consulting Editor, Nature Communications
Nature Portfolio

Heidelberger Platz 3, 14197 Berlin, Germany
Email: daryl.david@nature.com
ORCID: <https://orcid.org/0000-0002-9253-4805>

Reviewer #1:
Remarks to the Author:

In the revised version of the manuscript, the authors have addressed my concerns and suggestions. The paper could therefore be published in the current form.

Reviewer #2:

Remarks to the Author:

The authors have adequately addressed the concerns of this reviewer. This reviewer appreciates the time and energy the authors spent in 1) responding to this reviewer's critiques and 2) performing additional experiments to address this reviewer's concerns. This reviewer agrees with the authors that the inclusion of their model is important and warranted, as it is a first step towards the creation of a more comprehensive model and can be used to predict and understand any number of experimental outcomes in the future. The authors' brief discussion of their model is helpful for their goals and the results of the model. With the improvements made to the manuscript, this reviewer supports its publication in Nature Cell Biology.

Minor Comments:

- 1) If space allows a concluding / summary sentence in the abstract would be helpful.
- 2) In line 61, 'unstructured' should be 'disordered', as all proteins have some structure, whether primary, secondary, etc.
- 3) In line 282, 'truncates' should be 'truncations'.

Reviewer #3:

Remarks to the Author:

The revised manuscript addresses the issues I raised in the original review, and I believe it is suitable for publication. I have no further comments to add.

Author Rebuttal, first revision:

We thank the reviewers for all their helpful comments throughout the reviewing process. Our final answers are added below.

Reviewer #1:

Remarks to the Author:

In the revised version of the manuscript, the authors have addressed my concerns and suggestions. The paper could therefore be published in the current form.

No further comment

Reviewer #2:

Remarks to the Author:

The authors have adequately addressed the concerns of this reviewer. This reviewer appreciates the time and energy the authors spent in 1) responding to this reviewer's critiques and 2) performing additional experiments to address this reviewer's concerns. This reviewer agrees with the authors that the inclusion of their model is important and warranted, as it is a first step towards the creation of a more comprehensive model and can be used to predict and

understand any number of experimental outcomes in the future. The authors' brief discussion of their model is helpful for their goals and the results of the model. With the improvements made to the manuscript, this reviewer supports its publication in Nature Cell Biology.

Minor Comments:

1) If space allows a concluding / summary sentence in the abstract would be helpful.

Our current word count for the abstract is 147. Since the maximum is 150, we did not change the abstract

2) In line 61, 'unstructured' should be 'disordered', as all proteins have some structure, whether primary, secondary, etc.

We have made this change

3) In line 282, 'truncates' should be 'truncations'.

We have made this change

Reviewer #3:

Remarks to the Author:

The revised manuscript addresses the issues I raised in the original review, and I believe it is suitable for publication. I have no further comments to add.

No further comment

Final Decision Letter:

Dear Dr. Dogterom,

I am writing on behalf of my colleague, Dr. Daryl David, who has been out of the office.

We are pleased to inform you that your manuscript, "Multivalent interactions facilitate motor-dependent protein accumulation at growing microtubule plus ends", has now been accepted for publication in Nature Cell Biology.

Please note that Nature Cell Biology is a Transformative Journal (TJ). Authors may publish their research with us through the traditional subscription access route or make their paper immediately open access through payment of an article-processing charge (APC). Authors will not be required to make a final decision about access to their article until it has been accepted. Find out more about Transformative Journals

Authors may need to take specific actions to achieve compliance with funder and institutional open access mandates. If your research is supported by a funder that requires immediate open access (e.g. according to Plan S principles) then you should select the gold OA route, and we will direct you to the compliant route where possible. For authors selecting the subscription publication route, the journal's standard licensing terms will need to be accepted, including self-archiving policies. Those licensing terms will supersede any other terms that the author or any third party may assert apply to any version of the manuscript.

If you have not already done so, we strongly recommend that you upload the step-by-step protocols used in this manuscript to the Protocol Exchange (www.nature.com/protocolexchange), an open online resource established by Nature Protocols that allows researchers to share their detailed experimental know-how. All uploaded protocols are made freely available, assigned DOIs for ease of citation and are

fully searchable through nature.com. Protocols and Nature Portfolio journal papers in which they are used can be linked to one another, and this link is clearly and prominently visible in the online versions of both papers. Authors who performed the specific experiments can act as primary authors for the Protocol as they will be best placed to share the methodology details, but the Corresponding Author of the present research paper should be included as one of the authors. By uploading your Protocols to Protocol Exchange, you are enabling researchers to more readily reproduce or adapt the methodology you use, as well as increasing the visibility of your protocols and papers. You can also establish a dedicated page to collect your lab Protocols. Further information can be found at www.nature.com/protocolexchange/about

All the best,

Christina

Christina Kary, PhD

Chief Editor

Nature Cell Biology

1 New York Plaza

Tel: +44 (0) 207 843 4924
